# S-palmitoylation modulates ATG2-dependent non-vesicular lipid transport during starvation-induced autophagy

Wenhui Zheng [ID][1,4], Maomao Pu[1,4], Sai Zeng[1], Hongtao Zhang[1], Qian Wang[1], Tao Chen [ID][1], Tianhua Zhou [ID][1], Chunmei Chang [ID][2✉], Dante Neculai [ID][1✉] & Wei Liu [ID][1,3✉]

## Abstract

Lipid transfer proteins mediate the non-vesicular transport of lipids at membrane contact sites to regulate the lipid composition of organelle membranes. Despite significant recent advances in our understanding of the structural basis for lipid transfer, its functional regulation remains unclear. In this study, we report that S-palmitoylation modulates the cellular function of ATG2, a rod-like lipid transfer protein responsible for transporting phospholipids from the endoplasmic reticulum (ER) to phagophores during autophagosome formation. During starvation-induced autophagy, ATG2A undergoes depalmitoylation as the balance between ZDHHC11-mediated palmitoylation and APT1-mediated depalmitoylation. Inhibition of ATG2A depalmitoylation leads to impaired autophagosome formation and disrupted autophagic flux. Further, in cell and in vitro analyses demonstrate that S-palmitoylation at the C-terminus of ATG2A anchors the C-terminus to the ER. Depalmitoylation detaches the C-terminus from the ER membrane, enabling it to interact with phagophores and promoting their growth. These findings elucidate a S-palmitoylation-dependent regulatory mechanism of cellular ATG2, which may represent a broad regulatory strategy for lipid transport mediated by bridge-like transporters within cells.

**Keywords** ATG2; Autophagy; Lipid Transfer Protein; S-palmitoylation
**Subject Categories** Autophagy & Cell Death; Membranes & Trafficking; Post-translational Modifications & Proteolysis

## Introduction

Macroautophagy (hereafter referred to as autophagy) is a conserved lysosome-dependent degradation pathway in eukaryotes. The formation of autophagosomes, a hallmark of autophagy, involves the expansion and growth of the isolation membranes (phagophores) produced by the endoplasmic reticulum (ER). This process requires multiple autophagy-related proteins to work together. Among them, the large protein ATG2 (~2000 amino acids) containing the Chorein_N-domian plays a critical role in connecting the ER and phagophores by localizing to their contact sites (Gómez-Sánchez et al, 2018; Valverde et al, 2019; Osawa et al, 2019; Maeda et al, 2019). Studies on yeast have shown that Atg2 is essential for the effective formation of autophagosomes, and the lack of *atg2* leads to the obstruction of autophagy (Wang et al, 2001; Shintani et al, 2001). When the N-terminal of Atg2 binds to the ER, Atg2 uses its C-terminal to interact with the pre-autophagosomal structure (PAS) (Kotani et al 2018; Osawa et al, 2022).

In mammals, there are two functionally redundant ATG2 subtypes, ATG2A and ATG2B (Velikkakath et al, 2012). Simultaneous deletion of ATG2A and ATG2B results in blockage of autophagic flux and accumulation of unsealed premature phagophores (Velikkakath et al, 2012; Kishi-Itakura et al, 2014). Structural studies have shown that ATG2A and ATG2B form a rod-like structure and purified ATG2A can bridge liposomes (Chowdhury et al, 2018). Like the Chorein_N-domian-containing protein VPS13, which transfers phospholipids between the ER and other organelles (Kumar et al, 2018), ATG2 protein forms a long hydrophobic groove where the fatty acid chains of phospholipids can be accommodated, enabling ATG2 to transport phospholipids from the ER to phagophores for their expansion (Osawa et al, 2019; Valverde et al, 2019; Maeda et al, 2019). Recently, these long rod-like proteins, including ATG2 and VPS13, comprised of predicted anti-parallel β-strands-containing repeated modules, have been referred to as RBG proteins (Neuman et al, 2022). In addition, the ER-resident transmembrane proteins TMEM41B, VMP1, and phagophore protein ATG9 that interact with ATG2 have been identified as possessing scramblase activity, which promote ATG2-mediated lipid transfer flux by balancing the phospholipids molecules between phospholipid bilayer leaflets (Ghanbarpour et al, 2021; Matoba et al, 2020; Maeda et al, 2020).

Lipid modifications confer hydrophobicity on proteins and drives their association with membranes or specific membrane

[1]Department of Respiratory and Critical Care Medicine, Center for Metabolism Research, The Fourth Affiliated Hospital of Zhejiang University School of Medicine and International School of Medicine, International Institutes of Medicine, Zhejiang University, Yiwu, China. [2]Shanghai Key Laboratory of Metabolic Remodeling and Health, Institute of Metabolism and Integrative Biology, Fudan University, Shanghai, China. [3]Department of Ultrasound Medicine and State Key laboratory Implantation Device, The Second Affiliated Hospital of Zhejiang University, Hangzhou, China. [4]These authors contributed equally as first authors: Wenhui Zheng, Maomao Pu.
✉E-mail: chunmei_chang@fudan.edu.cn; dneculai@zju.edu.cn; liuwei666@zju.edu.cn

domains (Jiang et al, 2018; Mesquita et al, 2024). Among the known types of protein lipidation, S-acylation (frequently referred to as S-palmitoylation) is achieved by covalently linking palmitates to specific cysteine residues (Chamberlain and Shipston, 2015). This reversible process is catalyzed by a transmembrane family of palmitoyltransferases characterized by the presence of a zinc-finger aspartate-histidine-histidine-cysteine (DHHC) motif in their cysteine-rich domain (Salaun et al, 2010). There is growing evidence that in addition to mediating membrane attachment of soluble proteins, S-palmitoylation can occur in transmembrane proteins and regulate protein stability, protein-protein interactions, and protein trafficking (Wang et al, 2023; Fan et al, 2023; Zhang et al, 2020). Interestingly, recent studies have shown that S-palmitoylation is also involved in the regulation of autophagy. When DHHC13-mediated S-palmitoylation of ULK1 enhances its recruitment to autophagosome formation sites (Tabata et al, 2024), and DHHC5-mediated S-palmitoylation of Beclin-1 promotes class III phosphatidylinositol-3-kinase complex assembly (Guo et al, 2024), ZDHHC7-mediated S-palmitoylation of ATG16L1 facilitates LC3 lipidation during autophagy (Wei et al, 2024).

In this study, we report that ATG2A can undergo S-palmitoylation modification specifically regulated by palmitoyltransferase ZDHHC11 and acyl protein thioesterase 1 (APT1). By clarifying the necessity of depalmitoylation for the function of ATG2A in autophagosome formation, our findings reveal a previously unknown regulatory mechanism of lipid transfer proteins in cells.

# Results

## ATG2A undergoes depalmitoylation at its C-terminal during cell starvation

Research conducted on yeast has demonstrated that Atg2 serves as a phosphorylation substrate for Atg1, although it is unlikely that Atg1-dependent phosphorylation is directly related to the role of Atg2 in autophagy (Papinski et al, 2014). In our quest to identify novel post-translational modifications that may regulate the autophagic activity of ATG2, we performed an acyl-biotin exchange (ABE) assay on human ATG2. In this assay, thioester-linked acyl groups are substituted with biotin, which can be subsequently detected through Western blot analysis using horseradish peroxidase (HRP)-conjugated streptavidin. We found that both exogenous and endogenous ATG2A, as well as exogenous ATG2B, exhibited high levels of S-palmitoylation within cells (Figs. 1A–C and EV1A–C). The S-palmitoylation of ATG2A was effectively inhibited by 2-bromopalmitate (2-BP), a broad-spectrum inhibitor targeting ZDHHCs (Chen et al, 2017; Lu et al, 2019) (Figs. 1A–C and EV1A–C). Furthermore, intracellular Flag-ATG2A precipitated using anti-Flag beads displayed positive staining for 17-octadecynoic acid (17-ODYA), a bioorthogonal click chemistry probe for in situ labeling of palmitoyl protein (Martin et al, 2011; Lu et al, 2019) (Fig. 1D). These results indicated that intracellular ATG2 protein exists as an S-palmitoylated entity. Notably, culturing cells with starvation medium Earle's balanced salt solution (EBSS) led to a significantly reduction in ATG2 S-palmitoylation levels (Figs. 1A–C and EV1A–C). Consistent with this observation, removing glucose or serum from the culture

medium or treating cells with the mTOR inhibitor Torin1 also resulted in decreased S-palmitoylation levels of ATG2A (Figs. 1E and EV1D).

To further validate that ATG2 is an S-palmitoylated protein and undergoes depalmitoylation during cell starvation, we employed a combination of S-palmitoylation site prediction tools [https://swisspalm.epfl.ch/ and https://gpspalm.biocuckoo.cn (Ning et al, 2021)] along with sequence conservation analysis to identify the S-palmitoylation sites of ATG2A. The predictions indicate that potential sites are predominantly located at the C-terminal region of ATG2A (Table EV1), a finding corroborated by assessing the S-palmitoylation levels in both N-terminal (amino acids 1–846) and C-terminal (amino acids 847–1938) truncated forms of ATG2A (Fig. 1F). By generating and utilizing mutants of ATG2A, wherein the predicted cysteines were substituted with serine, we observed a significant reduction in the S-palmitoylation level following the alteration of conserved C-terminal C1704, C1713, and C1714 (Fig. EV1E–G). Notably, simultaneous mutations at these specific sites completely abolished the S-palmitoylation of ATG2A (Figs. 1G and EV1H), indicating that they represent key S-palmitoylation sites for this protein.

## ZDHHC11 mediates the S-palmitoylation of ATG2A

In mammals, the S-palmitoylation of proteins is primarily catalyzed by S-acyltransferases that contain a zinc-finger DHHC motif (ZDHHC), which comprises 23 members (Linder and Deschenes, 2007). To search for ATG2A-specific palmitoyltransferase, we conducted RNA interference (RNAi) targeting all 23 palmitoyltransferases in HEK293T cells. Remarkably, we observed that the levels of S-palmitoylation of ATG2A were significantly diminished only in cells deficient in ZDHHC11 (Figs. 2A and EV2A). Subsequently, we performed co-immunoprecipitation assays to investigate potential interactions between ATG2A and ZDHHC11. In cells expressing exogenous Flag-ATG2A alongside HA-ZDHHC11, immunoprecipitation of Flag-ATG2A resulted in the co-precipitation of HA-ZDHHC11 but not another ER-localized palmitoyltransferase, ZDHHC6 (Gorleku et al, 2011) (Fig. 2B). Furthermore, endogenous ZDHHC11 was found to be co-precipitated by endogenous ATG2A (Fig. EV2B), and in vitro pull-down assays using purified HA-ZDHHC11 and ATG2A indicated a direct interaction between these proteins (Fig. 2C). Moreover, confocal microscopy revealed that GFP-ATG2A can form small punctate structures with ER-localized Cherry-ZDHHC11 within cells (Figs. 2D and EV2C). Previous studies have indicated that ATG2A can localize on lipid droplets (LDs) (Velikkakath et al, 2012; Tamura et al, 2017). Consequently, we assessed the ATG2A/ZDHHC11 colocalization in relation to Lipi-Blue-labeled LDs. We observed colocalization between ATG2A and LDs, but no colocalizations were found between ATG2A/ZDHHC11 foci and LDs (Fig. EV2D). This suggests that ZDHHC11, a well-characterized quaternary ER transmembrane protein is unlikely to reside in phospholipid monolayers. ZDHHC11 is characterized by three fragments located on the cytoplasmic face of the ER membrane, including a long C-terminal tail. To verify and further characterize the interaction between ATG2A and ZDHHC11, we generated truncated mutants of ZDHHC11 by individually deleting each cytosolic domain. Co-immunoprecipitation assays revealed that the removal of the

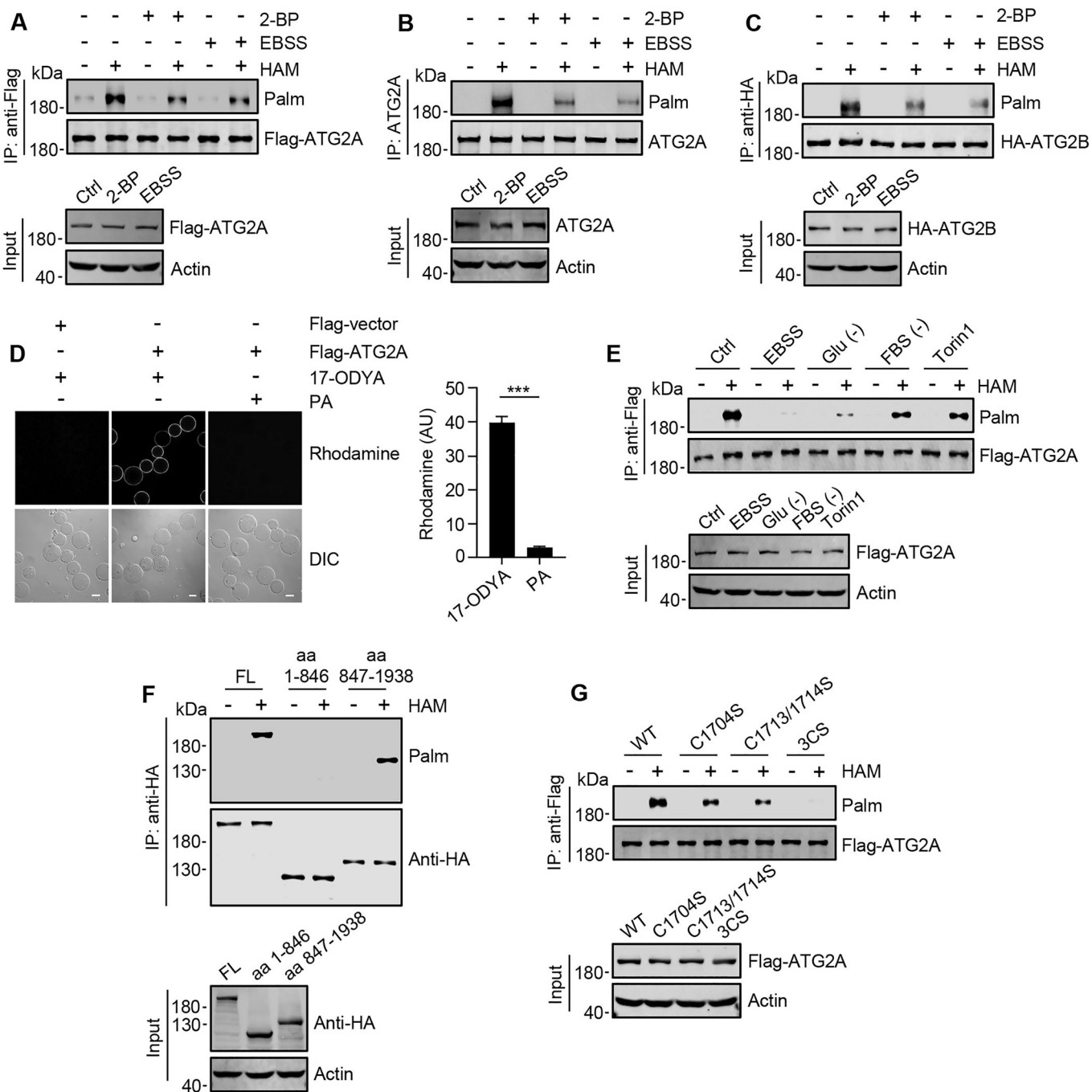

**Figure 1. ATG2 undergoes depalmitoylation at its C-terminal during cell starvation.**

(A–C) ATG2 S-palmitoylation in HEK293T cells. Cells were transfected with either Flag-ATG2A or Flag-ATG2B, or left untransfected, and treated with 2-BP or EBSS 48 h post-transfection. ABE assays analyzed the S-palmitoylation levels of Flag-ATG2A (A), endogenous ATG2A (B), and Flag-ATG2B (C). Hydroxylamine (HAM) cleaves and unmasks the thiol group of palmitoylated cysteine in ABE assays. 'Palm' indicates biotin levels on target protein detected via Streptavidin-HRP. (D) 17-ODYA metabolic labeling and click chemistry analysis of HEK293T cells expressing Flag-vector or Flag-ATG2A. Fluorescence and DIC microscopy imaged Flag-vector and Flag-ATG2A beads linked to TAMRA-azide using Cu(I). PA, palmitic acid. Scale bars, 50 μm. The data of statistical analysis is shown as mean ± SEM; $n = 30$ beads. Student's $t$-test was used to calculate $P$ value, ***$P < 0.001$. Exact $P$ value: $P = 6.30E{-}28$. (E) Detection of S-palmitoylation of Flag-ATG2A in HEK293T cells by ABE assay. Cells were cultured in EBSS, glucose starvation medium, serum starvation medium, or were treated with Torin1. (F) S-palmitoylation of the C-terminal or N-terminal truncated HA-ATG2A mutants in transfected HEK293T cells detected by ABE assay. (G) S-palmitoylation of Flag-ATG2A mutants assessed by ABE assay in HEK293T cells. 3CS: Cys 1704, Cys 1713, and Cys 1714 were substituted with serine. Source data are available online for this figure.

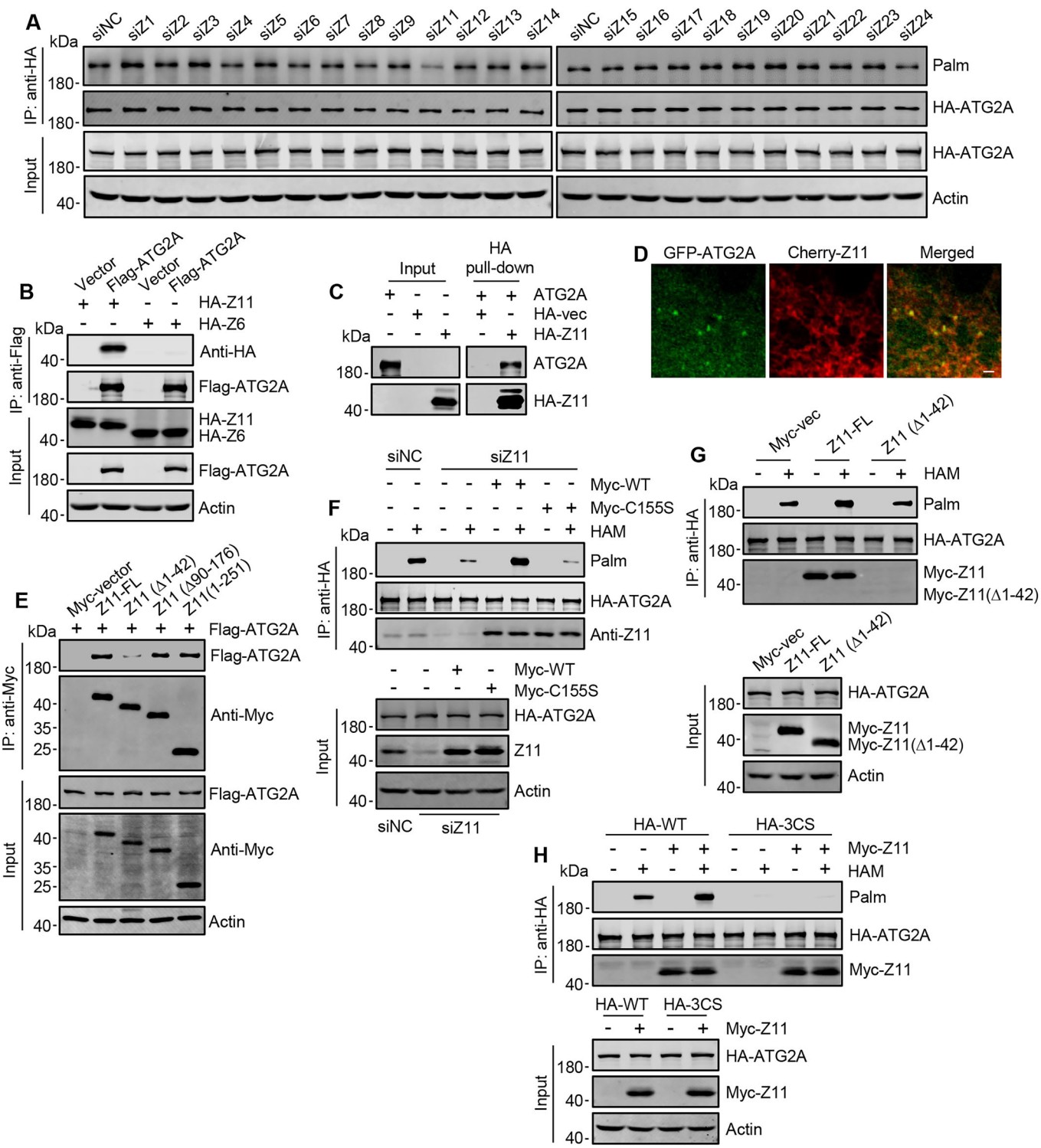

N-terminal cytosolic domain (amino acids 1–42) of ZDHHC11 completely abolished its interaction with ATG2A (Figs. 2E and EV2E).

We conducted knockdown-rescue experiments to establish that ZDHHC11 functions as the palmitoyltransferase for ATG2A. Notably, the knockdown of ZDHHC11 led to a reduction in S-palmitoylation of ATG2A, which could be reversed by re-

expressing wild-type ZDHHC11 but not the palmitoyltransferase-deficient variant ZDHHC11-C155S (Liu et al, 2018) (Figs. 2F and EV2F). Furthermore, overexpression of wild type ZDHHC11, rather than a truncated form lacking the N-terminal ATG2A-interaction domain (amino acid 1–42), enhanced ATG2A S-palmitoylation (Figs. 2G and EV2G). Additionally, overexpression of ZDHHC11 did not elevate S-palmitoylation levels in

**Figure 2. ZDHHC11 mediates the S-palmitoylation of ATG2A.**

(A) Screening of ZDHHCs for ATG2A S-palmitoylation. HEK293T cells expressing HA-ATG2A were transfected with specific siRNA targeting each of the ZDHHCs. ABE assays were performed 72 h after transfection. The siRNA for ZDHHC11 is designed to target ZDHHC11 and ZDHHC11B. (B) Co-immunoprecipitation of ZDHHC11 and ZDHHC6 with ATG2A. Flag-ATG2A was immunoprecipitated from HEK293T cells co-transfected with HA-ZDHHC11 or HA-ZDHHC6, and the precipitates were analyzed by Western blot using anti-HA. (C) In vitro pull-down assays detecting the direct binding of ZDHHC11 and ATG2A. Purified Flag-ATG2A was incubated with purified HA-ZDHHC11, and the precipitates were analyzed by Western blot using anti-ATG2A. (D) Co-localization of transiently expressed Cherry-ZDHHC11 and GFP-ATG2A in HeLa cells cultured in complete media. Scale bar, 1 μm. (E) Co-immunoprecipitation analysis of ATG2A and truncated ZDHHC11 mutants. Myc-ZDHHC11 and its truncated mutants were immunoprecipitated from HEK293T cells co-transfected with Flag-ATG2A. The precipitates were analyzed by Western blot using anti-Flag. (F) HA-ATG2A S-palmitoylation in HEK293T cells. Cells expressing HA-ATG2A were incubated with ZDHHC11 siRNA for 72 h. At the 24 h of incubation, Myc-ZDHHC11 or Myc-ZDHHC11-C155S was transfected. The cells were subjected to ABE assay 48 h after transfection. The cDNA of Myc-ZDHHC11 and Myc-ZDHHC11-C155S are siRNA-resistant. (G) ABE assay of HA-ATG2A S-palmitoylation in HEK293T cells with Myc-ZDHHC11 or Myc-ZDHHC11 (Δ1-42) expression. (H) S-palmitoylation of HA-ATG2A and HA-ATG2A-3CS in HEK293T cells with or without Myc-ZDHHC11 transfection. Source data are available online for this figure.

ATG2A with C-terminal cysteine substitutions at C1704, C1713, and C1714, indicating that these residues are the target sites for ZDHHC11 (Figs. 2H and EV2H). Collectively, these results suggest that ZDHHC11 serves as a palmitoyltransferase for ATG2A, facilitating S-palmitoylation at the C-terminus of ATG2A.

## APT1 is a depalmitoylase for ATG2A

To determine the depalmitoylase responsible for ATG2A, we initiated our investigation by examining the interacting proteins of ATG2A. HA-ATG2A was immunoprecipitated from HEK293T cells stably expressing HA-ATG2A, and the resulting precipitates were analyzed by mass spectrometry. The analysis revealed that acyl-protein thioesterases 1 and 2 (APT1 and APT2) are potential candidates (Dataset EV1). We confirmed the interaction between ATG2A and both APT1 and APT2 through co-immunoprecipitation assays (Fig. 3A). Furthermore, in vitro pull-down assays utilizing purified recombinant GST-APT1/APT2 alongside ATG2A corroborated the direct interaction between these proteins (Fig. 3B). Interestingly, while overexpression of either APT1 or APT2 led to a reduction in S-palmitoylation of ATG2A (Figs. 3C and EV3A), only downregulation of APT1 significantly enhanced ATG2A S-palmitoylation levels in cells (Figs. 3D and EV3B). Consistent with this observation, treatment with the specific APT1 inhibitor ML348 resulted in increased S-palmitoylation of ATG2A, whereas the specific APT2 inhibitor ML349 produced only a minimal effect (Fig. EV3C,D), indicating that depalmitoylation of ATG2A within cells is predominantly regulated by APT1. To further validate the role of APT1, we employed APT1 knockout cells. Notably, these knockout cells exhibited elevated levels of S-palmitoylated ATG2A compared to wild-type counterparts. Re-introduction of wild-type APT1 rather than an inactive mutant form lacking depalmitoylase activity (APT1-S119A) (Hirano et al, 2009; Huang et al, 2023) restored normal levels of S-palmitoylation for ATG2A (Figs. 3E and EV3E).

We investigated whether the observed depalmitoylation of ATG2A in starved cells was a consequence of APT1 activation or ZDHHC11 inactivation. Notably, when the knockout of APT1 increased the levels of ATG2A S-palmitoylation, EBSS treatment still caused a reduction in ATG2A S-palmitoylation in these cells (Figs. 3F and EV3F). Furthermore, an assessment of S-palmitoylation of ZDHHC11 itself, potentially indicative of its activity (Zhang et al, 2023), revealed that EBSS treatment did not significantly alter the S-palmitoylation levels of ZDHHC11 (Fig. EV3G,H). This was further corroborated by utilizing the

envelope protein from the Zika virus (ZIKV) as a known substrate for ZDHHC11 (Hu et al, 2023), demonstrating no discernible difference in S-palmitoylation between transfected ZIKV envelope proteins from starved and non-starved cells (Fig. EV3I,J). Meanwhile, EBSS treatment did not result in detectable alterations in the cytosolic distribution of APT1 (Fig. EV3K) and the interaction between ATG2A and APT1 (Fig. EV3L). Finally, we examined the interaction between ATG2A and ZDHHC11 under cell starvation conditions. The results indicated that compared to untreated cells, EBSS treatment significantly reduced the co-precipitation levels of ZDHHC11 with ATG2A in cells with or without APT1 deletion (Fig. 3G,H). These data suggest that the observed reduction in S-palmitoylation levels of ATG2A during cell starvation is not attributable to changes in the intrinsic activity of ZDHHC11 protein or the activation of APT1. Rather, it appears to stem from diminished interactions between ATG2A and ZDHHC11.

## Depalmitoylation of ATG2A is essential for starvation-induced autophagy

To investigate the role of ATG2A depalmitoylation in autophagy, we first assessed the degradation of autophagic substrate p62. In ZDHHC11 knockdown cells, we observed that the levels of p62 protein rather than p62 mRNA was significantly reduced, which could be prevented by the lysosomal inhibitor chloroquine (Figs. 4A,B and EV4A), indicating that the absence of ZDHHC11 enhances autophagic degradation of p62. In addition, overexpression of ZDHHC11, as opposed to the inactive variant ZDHHC11-C155S, effectively suppressed starvation-induced p62 degradation in cells stably expressing HA-ATG2A instead of unpalmitoylatable HA-ATG2A-3CS (Figs. 4C and EV4B). These results suggest that S-palmitoylation of ATG2A exerts an inhibitory effect on autophagy. Subsequently, utilizing ATG2A/2B double-knockout (ATG2A/2B-DKO) NRK cells, we established stable cell lines expressing HA-ATG2A or HA-ATG2A-3CS devoid of endogenous ATG2A and ATG2B. Consistent with previous observations (Velikkakath et al, 2012), deletion of both ATG2A and ATG2B led to the formation of large GFP-LC3B puncta alongside a reduced number of normal-sized GFP-LC3B dots in starved cells (Fig. 4D,E). When this phenotype was fully corrected by introducing either HA-ATG2A or HA-ATG2A-3CS, overexpression of ZDHHC11 counteracted the effects of HA-ATG2A but not HA-ATG2A-3CS (Fig. 4D,E). Correspondingly, the S-palmitoylation levels in cells expressing HA-ATG2A but not HA-ATG2A-3CS were increased by ZDHHC11 overexpression even under starvation conditions

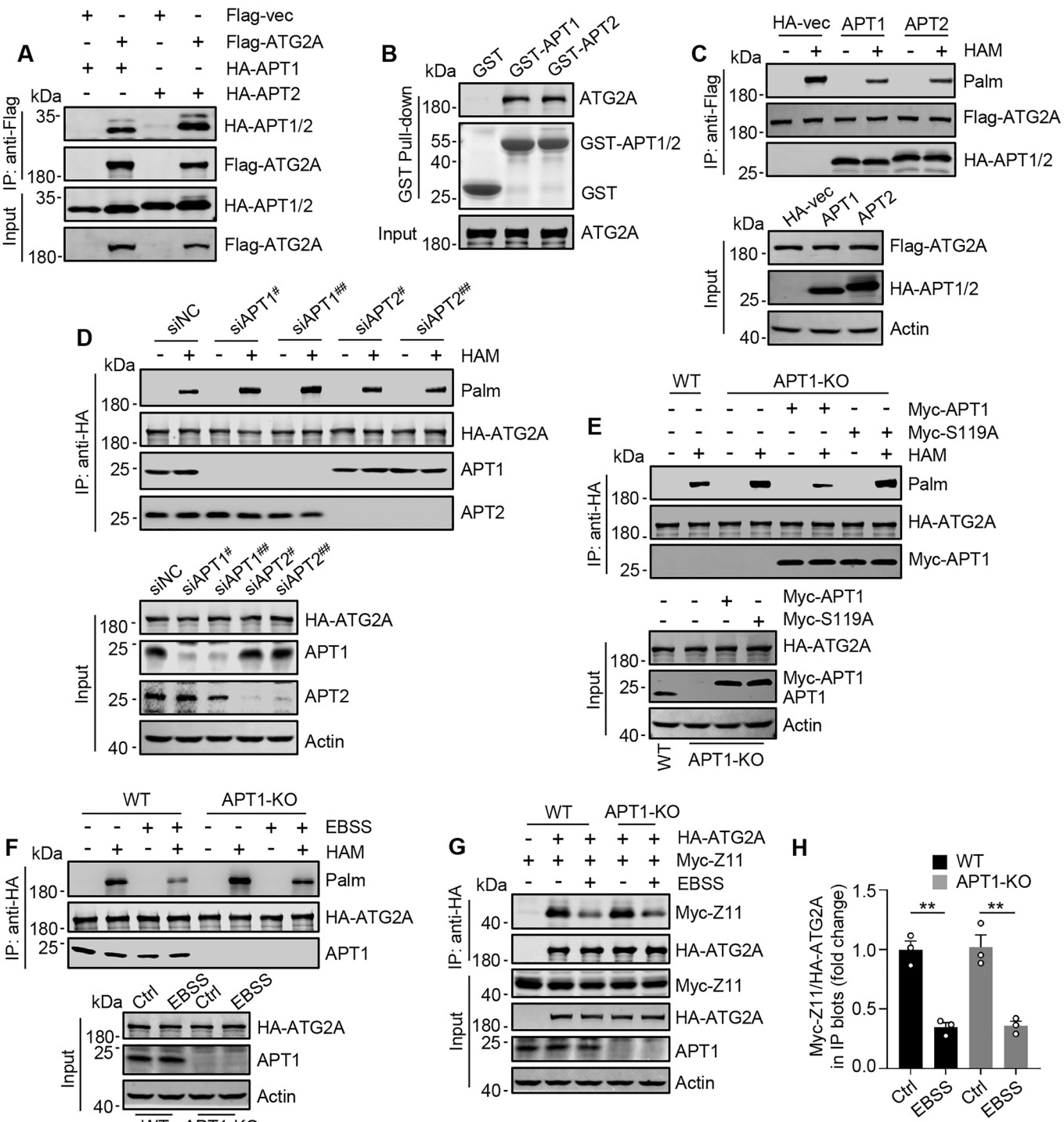

(Fig. EV4C). These results suggest that inhibiting depalmitoylation of ATG2 can mimic the consequences observed upon loss of ATG2, leading to impaired autophagosome formation (Velikkakath et al, 2012). Subsequently, we assessed autophagy flux utilizing cell systems that express Cherry-GFP-LC3B. The results indicated that in ATG2A/2B-DKO cells, the starvation-induced increase in the ratio of red-to-total puncta indicative enhanced autophagy flux, was abolished (Fig. 4F,G). When re-introduction of HA-ATG2A or

HA-ATG2A-3CS into these cells restored the ratio, overexpression of ZDHHC11 inhibited the effect of HA-ATG2A but not that of HA-ATG2A-3CS (Fig. 4F,G). Further evaluation of autophagy flux was conducted through HaloTag processing assays (Yim et al, 2022), which utilize the resistance of ligand-bound Halo to lysosomal degradation. Consequently, the quantity of ligand-Halo generated from the lysosomal degradation of ligand-Halo-LC3B serves as an indicator for levels of autophagy flux. We observed that

◄ **Figure 3. APT1 is a depalmitoylase for ATG2A.**

(A) Interaction of APT1 and APT2 with ATG2A. Flag-ATG2A was immunoprecipitated from HEK293T cells cultured in complete media and transfected with HA-APT1 or HA-APT2, and then the precipitates were analyzed by Western blot using anti-HA. (B) Pull-down assay of APT1/2-ATG2A binding. Purified Flag-ATG2A was incubated with recombinant GST-APT1 or GST-APT2, and the bound ATG2A was detected by Western blot using anti-ATG2A. (C) Flag-ATG2A S-palmitoylation in HEK293T cells transfected with HA-APT1 or HA-APT2. ABE assay was performed 48 h after transfection. (D) S-palmitoylation of HA-ATG2A in HEK293T cells expressing HA-ATG2A. The cells were incubated with APT siRNAs for 72 h and ABE assay was conducted. (E) HA-ATG2A S-palmitoylation in APT1-KO HEK293T cells expressing HA-ATG2A. The cells were transfected with Myc-APT1 or Myc-APT1-S119A. ABE assay was carried out 48 h after transfection. (F) S-palmitoylation of HA-ATG2A in WT and APT1-KO HEK293T cells expressing HA-ATG2A. The cells were treated with or without EBSS. (G) Co-immunoprecipitation of ZDHHC11 with ATG2A. HA-ATG2A was immunoprecipitated from HA-ATG2A-expressing WT or APT1-KO HEK293T cells transfected with Myc-ZDHHC11 and treated with or without EBSS. The precipitates were analyzed by Western blot using anti-Myc. (H) Statistical analysis of (G). The data are presented as mean ± SEM of three independent experiments. Student's *t*-test was used to calculate *P* values, **$P < 0.01$. Exact *P* values from left to right: $P = 0.00137$; $P = 0.00371$. Source data are available online for this figure.

expression of either HA-ATG2A or HA-ATG2A-3CS in ATG2A/2B-DKO cells reinstated starvation-triggered generation of free Halo; however, overexpression of ZDHHC11 impeded the effect of HA-ATG2A while leaving that of HA-ATG2A-3CS unaffected (Figs. 4H and EV4D).

We investigated the role of APT1 in starvation-induced autophagy. Our findings indicated that in the absence of APT1, EBSS treatment still led to the degradation of p62, which could be mitigated by the overexpression of ZDHHC11 (Figs. 4I and EV4E). Notably, the overexpression of APT1 did not further enhance p62 degradation triggered by EBSS treatment, and co-expression of ZDHHC11 was able to attenuate p62 degradation in these cells (Figs. 4I and EV4E). Consistent with these results, in GFP-LC3B-expressing cells subjected to EBSS treatment, the overexpression of APT1 was found to mitigate the formation of large GFP-LC3B puncta induced by ZDHHC11 overexpression when compared to wild-type cells. The knockdown of APT1 resulted in a mild increase in the generation of large GFP-LC3B puncta, an effect that could be further amplified by overexpression of ZDHHC11 (Figs. 4J and EV4F).

It is important to note that in all conducted experiments, we did not observe any significant alterations in the expression levels of ATG2A in starved cells or in cells overexpressing ZDHHC11 or APT1. This suggests that S-palmitoylation may not influence the stability of the ATG2A protein.

## S-palmitoylation anchors the C-terminal of ATG2A to the ER membrane

To explore the molecular mechanism by which S-palmitoylation modulates ATG2 function, we initially evaluated the potential influence of S-palmitoylation on the lipid transfer activity of ATG2A through in vitro lipid transfer assays (Valverde et al, 2019; Maeda et al, 2019; van Vliet et al, 2022). The donor liposomes containing NBD-phosphatidylethanolamine (NBD-PE) and Rhodamine-phosphatidylethanolamine (Rh-PE) were combined with receptor liposomes that lacked fluorescent lipids. The presence of rhodamine on the donor liposomes quenches NBD fluorescence, thereby inhibiting emission from NBD. When either NBD-PE or Rh-PE is transferred to recipient liposomes, an increase in NBD fluorescence is observed. Our findings indicated that the addition of purified Flag-ATG2A or Flag-ATG2A-3CS to the liposome mixture resulted in an equal enhancement in NBD fluorescence intensity (Fig. 5A,B). Given that ATG2 does not facilitate liposome fusion (Valverde et al, 2019), these results suggest that ATG2A and ATG2A-3CS exhibit equivalent lipid transfer activity.

Then, we investigated whether depalmitoylation influences the binding of ATG2 to ER membranes, which is primarily mediated by its N-terminal region (Kotani et al, 2018). Subcellular fractionation was performed on ATG2A/2B-DKO cells expressing wild-type ATG2A, ATG2A-3CS or mutants lacking the amphipathic helix located in the C-terminus that is essential for phagophore binding (ATG2A-ΔAH and ATG2A-3CS-ΔAH) (Kotani et al, 2018; Tamura et al, 2017). Our findings revealed that all these ATG2A variants were comparably distributed across ER fractions (Fig. EV5A,B), and cell imaging showed co-localizations of Cherry-tagged ATG2A-WT and ATG2A-3CS with GFP-Sec61β (Fig. EV5C,D). Furthermore, co-immunoprecipitation assays indicated that the interaction between both ATG2A/ATG2A-3CS and VMP1 remained consistent in both starved and non-starved cells (Fig. EV5E). These results suggest that S-palmitoylation of ATG2A does not impact its N-terminal-mediated association with the ER.

The enrichment of S-palmitoylation sites at the C-terminal region of ATG2A suggests that S-palmitoylation may facilitate the anchoring of the C-terminus to the ER membrane. To investigate this hypothesis, we generated a truncated mutant comprising only the C-terminal region of ATG2A (amino acids 847–1938) and lacking the amphipathic helix (ATG2A-CT-ΔAH). Upon expression in ATG2A/2B-DKO cells, cell fractionation assays revealed a substantial presence of ATG2A-CT-ΔAH in the ER fraction, while significant less ATG2A-CT-3CS-ΔAH was detected in the fraction (Fig. 5C,D). These data indicate that the C-terminal region of ATG2A is capable of binding to ER, and this interaction is dependent on S-palmitoylation. This conclusion was further supported by co-localization observations showing that Cherry-ATG2A-CT colocalized with GFP-Sec61β in cells, whereas Cherry-ATG2A-CT-3CS did not exhibited such co-localization (Fig. 5E). Additionally, split super-positive GFP (split-spGFP) assay (Yang et al, 2018) indicated that only spGFP$_{11}$ fused with ATG2A-CT could form fluorescent GFP when paired with spGFP$_{1-10}$ anchored to ER; spGFP$_{11}$ linked to ATG2A-CT-3CS failed to produce fluorescence under similar conditions (Fig. 5F,G). Collectively, these results suggest that S-palmitoylation at the C-terminal region of ATG2A plays a critical role in anchoring this domain to the ER membrane.

To verify that the binding of the C-terminal region of ATG2A to the ER membrane can inhibit autophagy, we generated a mutant of ATG2A by incorporating an ER targeting sequence (Tiwari and Weissman, 2001; Yang et al, 2018) at its C-terminus (designated as ATG2A-ER) and confirmed that this mutant was fully localized to the ER (Fig. EV5F). When expressed in ATG2A/2B-DKO cells, under starvation conditions, this mutant did not reduce the

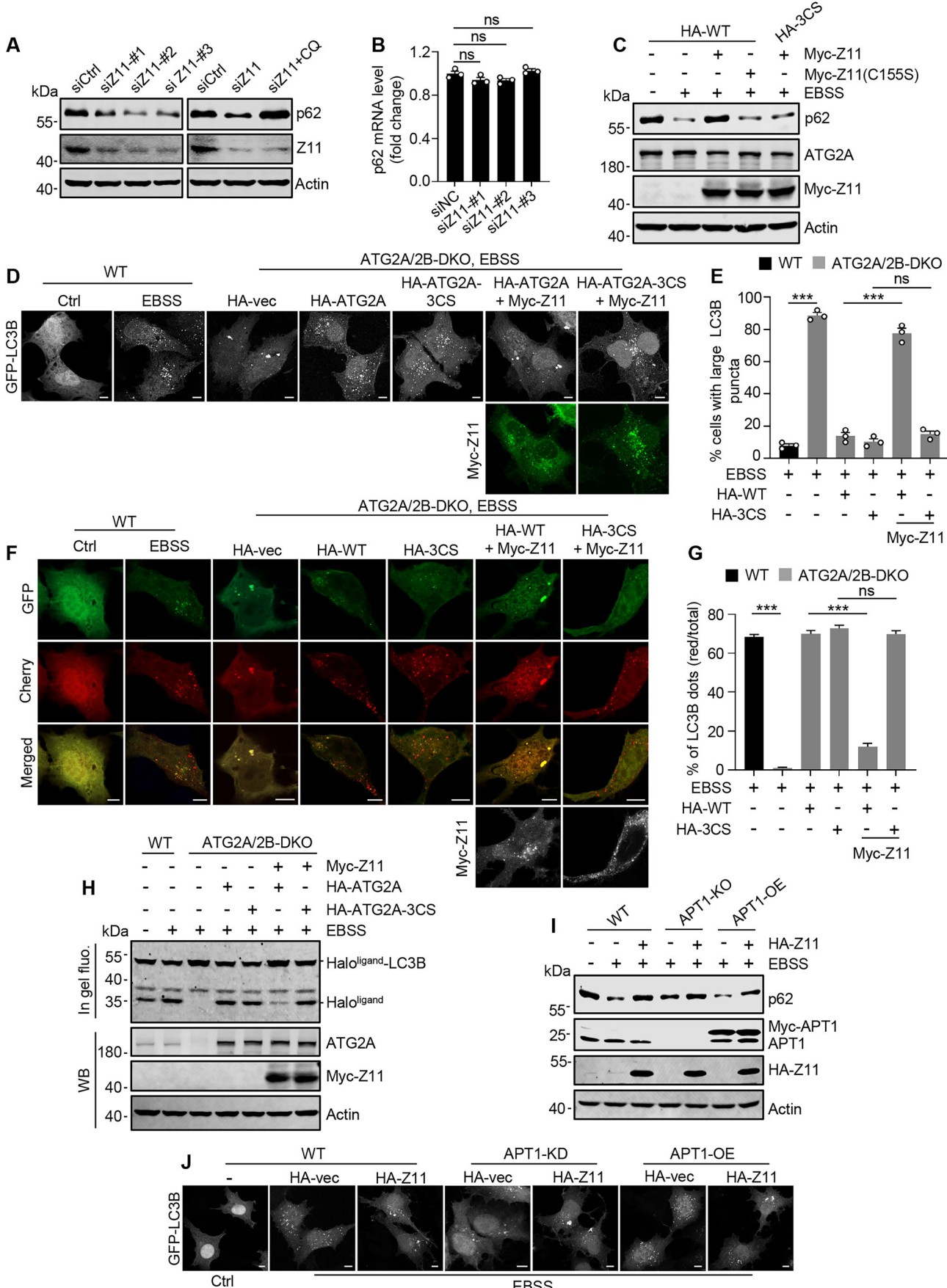

Figure 4. Depalmitoylation of ATG2A is essential for starvation-induced autophagy.

(A) p62 protein levels in HEK293T cells transfected with ZDHHC11 siRNA and treated with or without chloroquine (CQ). (B) Statistical analysis of p62 mRNA levels in HEK293T cells transfected with ZDHHC11 siRNA. (C) p62 proteins levels in HEK293T cells stably expressing HA-ATG2A or HA-ATG2A-3CS. The cells were transfected with Myc-ZDHHC11 or Myc-ZDHHC11-C155S and cultured in EBSS. (D) Representative images of GFP-LC3B in WT or ATG2A/B-DKO NRK cells stably expressing HA-ATG2A or HA-ATG2A-3CS. The cells were transfected with GFP-LC3B and with or without Myc-ZDHHC11. Cells were then cultured in complete media or in EBSS for 4 h. Scale bars, 5 μm. (E) Statistical analysis of the proportion of cells containing large LC3B puncta (diameter > 1.5 μm) in (D). Student's $t$-test was used to calculate $P$ values. Exact $P$ values from left to right: $P = 2.71E−06$; $P = 6.92E−05$; $P = 0.13493$. (F) Representative images of Cherry-GFP-LC3B in WT or ATG2A/B-DKO NRK cells stably expressing HA-ATG2A or HA-ATG2A-3CS. The cells were transfected with Cherry-GFP-LC3B and with or without Myc-ZDHHC11. Cells were then cultured in complete media or in EBSS for 4 h. Scale bars, 5 μm. (G) Statistical analysis of (F). $n = 30$ cells. Student's $t$-test was used to calculate $P$ values. Exact $P$ values from left to right: $P = 1.41E−48$; $P = 1.26E−31$; $P = 0.20291$. (H) HaloTag Processing assays for evaluating autophagy flux in WT or ATG2A/B-DKO NRK cells stably expressing HA-ATG2A or HA-ATG2A-3CS. The cells were transfected with Halo-LC3B and with or without Myc-ZDHHC11. 48 h post-transfection, cells were pulsed with tetramethylrhodamine (TMR)-conjugated ligands for 20 min in complete media and then cultured in complete media or EBSS for 4 h. (I) p62 protein levels in WT, APT1-KO and APT1-overexpression (APT1-OE) HEK293T cells transfected with or without HA-ZDHHC11 and then treated with or without EBSS. (J) Representative images of GFP-LC3B in WT, APT1-knockdown (APT1-KD) and APT1-overexpression (APT1-OE) NRK cells transfected with or without HA-ZDHHC11 and then treated with or without EBSS. Scale bars, 5 μm. Data information: All statistical data are presented as mean ± SEM of three independent experiments unless otherwise specified. ns, not significant; ***$P < 0.001$ (Student's $t$-test). Source data are available online for this figure.

accumulation of p62 induced by ATG2A/2B-DKO, in contrast to wild-type ATG2A or ATG2A-3CS (Fig. 5H,I). In addition, in comparison with cells expressing GFP-ATG2A, those expressing GFP-ATG2A-ER exhibited an increased formation of large Cherry-LC3B puncta and a decrease in normal-sized Cherry-LC3B dots (Fig. 5J,K). These results indicate that anchoring the C-terminus of ATG2A to the ER indeed inhibits starvation-induced autophagy.

## Depalmitoylation is necessary for ATG2A's interaction with phagophores

The C-terminus of ATG2 is responsible for targeting ATG2 to forming autophagosomes. Therefore, we investigated the effect of S-palmitoylation on the binding affinity of ATG2A to autophagic membranes within cells. Myc-ZDHHC11 was co-transfected with GFP-LC3B into ATG2A/2B-DKO cells that stably express HA-ATG2A or HA-ATG2A-3CS. Following cell starvation, GFP-LC3B positive membranes were isolated using GFP-TRAP magnetic beads and subsequently analyzed via Western blot (Gao et al, 2010; Pu et al, 2023). We found that, despite comparable levels of GFP-LC3B-II and ATG5-ATG12, a greater amount of HA-ATG2A-3CS was associated with GFP-LC3B membranes compared to HA-ATG2A (Fig. 6A,B). Consistently, comparing cells expressing HA-ATG2A, which exhibited fewer GFP-LC3B dots and limited localization on GFP-LC3B, cells expressing HA-ATG2A-3CS displayed an increased number of GFP-LC3B dots along with enhanced colocalization between HA-ATG2A-3CS and GFP-LC3B (Fig. 6C,D). Using GFP-FIP200 as a marker for phagophores yielded similar results to obtained using GFP-LC3B (Fig. 6E,F). Collectively, these results indicate that depalmitoylation facilitates the interaction between ATG2A and autophagic membranes.

Next, we investigated the impact of S-palmitoylation on the interaction between ATG2A and ATG9A, which is localized in the phagophore during autophagosome formation (Broadbent et al, 2023; Olivas et al, 2023). Under basal conditions, immunoprecipitation analysis revealed that ATG2A-3CS co-precipitated a greater amount of ATG9A compared to wild-type ATG2A (Fig. 6G). Furthermore, cell starvation enhanced the co-precipitation of ATG9A with ATG2A; however, it did not further augment the co-precipitation of ATG9A with ATG2A-3CS (Fig. 6G). In contrast, when employing the mutant form of ATG2A (ATG2A-mYFS), which does not interact with WIPI4 (Zheng et al,

2017), as a control, WIPI4 was found to be equally co-precipitated by both ATG2A and ATG2A-3CS (Fig. 6H). These results suggest that S-palmitoylation modulates the interaction between ATG2A and ATG9A but does not influence its capacity to form complexes with WIPI4.

In addition to transferring lipids from the ER to forming autophagosomes, recent studies have revealed that the lipid transfer activity of ATG2 also plays a role in the rapid repair of lysosomal damage (Tan and Finkel, 2022; Cross et al, 2023). Consequently, we investigated the impact of S-palmitoylation of ATG2A in lysosomal damage repair. Notably, treatment with the lysomotropic agent L-Leucyl-L-Leucine methyl ester (LLOMe) induced a time-dependent decrease in S-palmitoylation levels of ATG2A (Fig. 6I,J), suggesting that this modification may be involved in lysosomal repair mechanisms. Subsequently, we transfected GFP-galectin-3 into ATG2A/2B-KO cells stably expressing HA-ATG2A or HA-ATG2A-3CS to label damaged lysosomes (Aits et al, 2015). Although LLOMe treatment led to comparable levels of GFP-galectin-3 puncta in both cell types, indicating similar lysosomal damage, in cells expressing HA-ATG2A but not those expressing HA-ATG2A-3CS, overexpression of Myc-ZDHHC11 resulted in a significant increase in the number of GFP-galectin-3 puncta following LLOMe treatment (Fig. 6K,L), suggesting that depalmitoylation of ATG2A indeed contributes to the repair process for damaged lysosomes.

## Discussion

In this study, we have identified that the lipid transfer protein ATG2A is an S-palmitoylated protein that undergoes depalmitoylation in cells under starvation conditions. Our findings indicate that ATG2A necessitates a depalmitoylation process to fulfill its role in autophagy. We propose that this dynamic protein-membrane binding mechanism may also extend to the functional regulation of other rod-like lipid transfer proteins within cells, given the significant structural similarities they share. The S-palmitoylation-dependent reversible binding of one terminal to the donor membrane function as a drawbridge, modulating its interaction with the recipient membrane and thereby controlling its lipid transfer process (Fig. 6M).

The structural basis of ATG2-mediated lipid transfer and the membrane binding domains of ATG2 have been extensively investigated

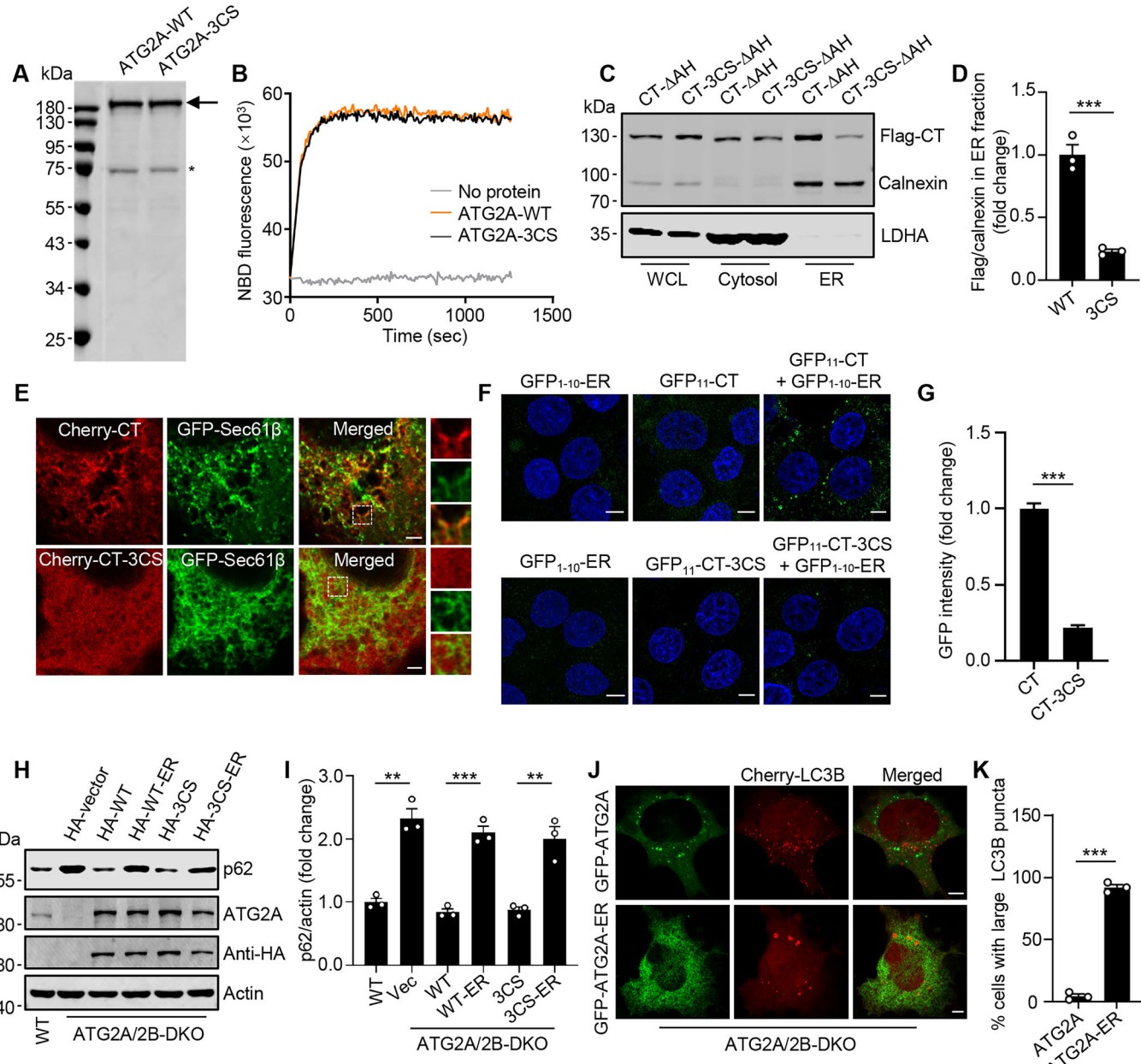

**Figure 5. S-palmitoylation anchors the C-terminal of ATG2A to the ER membrane.**

(A) Coomassie blue staining of purified Flag-ATG2A and Flag-ATG2A-3CS. Arrow indicates the band of ATG2A proteins. Asterisk indicates non-specific bands. (B) In vitro lipid transfer assay of Flag-ATG2A and Flag-ATG2A-3CS using liposomes. The data are representative from one of three independent experiments. (C) Western blot analysis of the distribution of the C-terminal of ATG2A in subcellular fractions. ATG2A/2B-DKO NRK cells were transfected with Flag-ATG2A-CT-ΔAH and Flag-ATG2A-CT-3CS-ΔAH. The cells were fractionated and analyzed 48 h after transfection. WCL: whole cell lysate. (D) Statistical analysis of (C). Student's *t*-test was used to calculate *P* value. Exact *P* value: *P* = 0.00072. (E) Representative images showing co-localization of Cherry-ATG2A-CT or Cherry-ATG2A-CT-3CS with GFP-Sec61β. Cherry-ATG2A-CT or Cherry-ATG2A-CT-3CS was co-transfected with GFP-Sec61β in HeLa cells and the cells were imaged 48 h post-transfection. Scale bars, 2 μm. (F) split-spGFP assay in HeLa cells. Cells were transfected with indicated fusion proteins and GFP signal were detected by confocal microscopy. Scale bars, 5 μm. (G) Statistical analysis of (F). *n* = 30 cells. Student's *t*-test was used to calculate *P* value. Exact *P* value: *P* = 6.31E−27. (H) p62 proteins levels in ATG2A/2B-DKO NRK cells expressing indicated fusion proteins and treated with EBSS for 4 h. (I) Statistical analysis of p62 protein levels in (H). Student's *t*-test was used to calculate *P* values. Exact *P* values from left to right: *P* = 0.00122; *P* = 0.00047; *P* = 0.00443. (J) Representative images of transfected Cherry-LC3B in ATG2A/B-DKO NRK cells expressing GFP-ATG2A or GFP-ATG2A-ER. The cells were cultured with EBSS for 4 h. Scale bars, 5 μm. (K) Statistical analysis of the proportion of cells containing large LC3B puncta (diameter > 1.5 μm) in (J). Student's *t*-test was used to calculate *P* value. Exact *P* value: *P* = 6.72E−06. Data information: All statistical data are presented as mean ± SEM of three independent experiments unless otherwise specified. **P < 0.01; ***P < 0.001 (Student's *t*-test). Source data are available online for this figure.

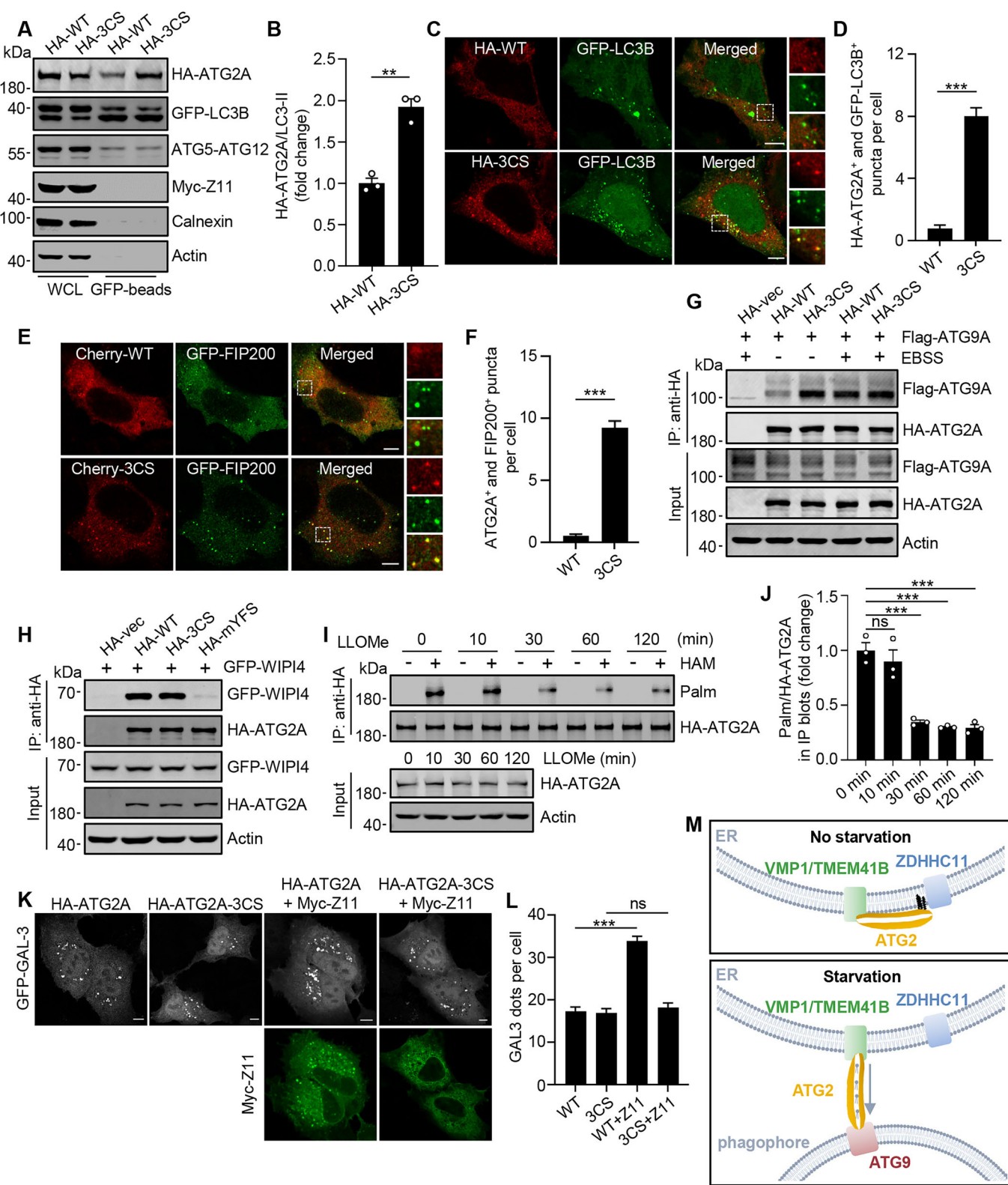

**Figure 6.  Depalmitoylation is necessary for ATG2A's interaction with phagophores.**

(**A**) Western blot analysis of HA-ATG2A bound to GFP-LC3B membranes from ATG2A/2B-DKO NRK cells stably expressing HA-ATG2A or HA-ATG2A-3CS. The cells were transfected with GFP-LC3B and Myc-ZDHHC11 and treated with EBSS for 4 h. GFP-LC3B membranes were isolated by affinity purification using GFP-TRAP magnetic beads. (**B**) Statistical analysis of (A). Student's *t*-test was used to calculate *P* value. Exact *P* value: *P* = 0.00121. (**C**) Representative images showing colocalizations of HA-ATG2A and HA-ATG2A-3CS with GFP-LC3B. ATG2A/2B-DKO NRK cells stably expressing HA-ATG2A or HA-ATG2A-3CS were co-transfected with GFP-LC3B and Myc-ZDHHC11 and cultured in EBSS for 4 h. Scale bars, 5 μm. (**D**) Statistical analysis of (C). *n* = 30 cells. Student's *t*-test was used to calculate *P* value. Exact *P* value: *P* = 1.15E −17. (**E**) Representative images displaying the colocalization of Cherry-ATG2A and Cherry-ATG2A-3CS with GFP-FIP200 in HeLa cells that stably express Myc-ZDHHC11. The cells were treated with EBSS. Scale bars, 5 μm. (**F**) Statistical analysis of (E). *n* = 30 cells. Student's *t*-test was used to calculate *P* value. Exact *P* value: *P* = 8.97E−23. (**G**) Co-immunoprecipitation of ATG9A with ATG2A or ATG2A-3CS. HA-ATG2A and HA-ATG2A-3CS were immunoprecipitated from HEK293T cells co-transfected with Flag-ATG9A. The cells were treated with or without EBSS, and the precipitates were analyzed by Western blot using anti-Flag. (**H**) Co-immunoprecipitation of WIPI4 with ATG2A. HA-ATG2A, HA-ATG2A-3CS, or HA-ATG2A-myFS was immunoprecipitated from HEK293T cells cultured in complete media and co-transfected with GFP-WIPI4. The precipitates were analyzed by Western blot using anti-GFP. (**I**) S-palmitoylation of HA-ATG2A detected by ABE assay in HEK293T cells treated with 1 mM LLOMe for indicated times. (**J**) Statistical analysis of S-palmitoylation levels for HA-ATG2A in (I). Student's *t*-test was used to calculate *P* values. Exact *P* values from left to right: *P* = 0.46999; *P* = 0.00088; *P* = 0.00061; *P* = 0.00079. (**K**) Representative images showing GFP-galectin-3 puncta in ATG2A/2B-DKO NRK cells stably expressing HA-ATG2A or HA-ATG2A-3CS. The cells were co-transfected with GFP-galectin-3 and Myc-ZDHHC11, and were treated with 1 mM LLOMe for 2 h. Scale bars, 5 μm. (**L**) Statistical analysis of (K). *n* = 30 cells. Student's *t*-test was used to calculate *P* values. Exact *P* values from left to right: *P* = 2.85E−15; *P* = 0.40442. (**M**) Schematic Model of ATG2A functioning by its C-terminal depalmitoylation. Under basal conditions, S-palmitoylation of the C-terminal of ATG2A by ZDHHC11 anchors it onto the ER. Upon cell starvation, depalmitoylation allows the C-terminus to be separated from ER membrane and to interact with phagophores. Data information: All statistical data are presented as mean ± SEM of three independent experiments unless otherwise specified. ns, not significant; **P* < 0.01; ***P* < 0.001 (Student's *t*-test). Source data are available online for this figure.

(Osawa et al, 2022; Tamura et al, 2017; Kotani et al, 2018; Chowdhury et al, 2018). While both the N-terminal and C-terminal regions of ATG2 exhibit affinity for membranes, studies conducted in cellular contexts suggest that the function of the N-terminal region is linked to its interaction with the ER, whereas the C-terminal amphipathic helix plays a crucial role in directing ATG2A to phagophores and lipid droplets (Tamura et al, 2017; Kotani et al, 2018). Our findings revealed a novel mode of binding between ATG2 and membranes, which can dynamically regulate the physiological function of ATG2. The S-palmitoylation sites we identified are located at the C-terminal region of ATG2A, adjacent to the amphipathic helix. This configuration permits the C-terminus to interact with the ER membrane, thereby preventing contact between the amphipathic helix and autophagic membranes while maintaining its association with the ER through its N-terminus. The variability in spatial positioning renders this bridge-like protein a flexible and controllable drawbridge. Supporting this notion, S-palmitoylation specifically influences interactions between ATG2A and ATG9, a phagophore transmembrane protein, while not affecting interactions with WIPI4, a cytosolic protein that forms complexes with ATG2A independently of phagophore formation (Obara et al, 2008).

Our RNAi screening and in cell/in vitro analyses demonstrate that ER-localized ZDHHC11 serves as a selective palmitoyltransferase for ATG2A. In contrast to ZDHHC5, ZDHHC7, and ZDHHC13, which primarily target the Golgi apparatus and promote autophagy (Guo et al, 2024; Wei et al, 2024, Tabata et al, 2024), our results indicate that ZDHHC11 negatively regulates starvation-induced autophagy. In addition, while ZDHHC19 is also localized to the ER, the S-palmitoylation of autophagy receptor p62 mediated by ZDHHC19 remains unaffected under starvation conditions (Huang et al, 2023). Although our understanding of the regulatory mechanisms governing the activity of ZDHHCs within cells is currently limited, these data suggest that cellular activation of ZDHHCs may not be primarily dependent on intracellular levels of palmitoyl-CoA. Consistent with this notion, our results reveal that the activity of ZDHHC11 remains unchanged in starved cells. The observed decrease in ATG2A S-palmitoylation is attributed to reduced interaction between ATG2A and ZDHHC11 during starvation. Currently, the precise mechanism underlying the reduced interaction between ATG2A and ZDHHC11

in starved cells remains unclear. Given the significant decrease in S-palmitoylation of ATG2 induced by glucose and serum deprivation, as well as Torin 1 treatment, it is plausible that alterations in modifications to either ZDHHC11 or ATG2A mediated directly or indirectly by mTOR may play a role in this process.

Autophagy is a complex multistep process that encompasses initiation, vesicle nucleation, membrane elongation and closure, as well as degradation stages. In conjunction with the identification of S-palmitoylation in other autophagy-related proteins involved in vesicle nucleation (Guo et al, 2024; Wei et al, 2024; Tabata et al, 2024), our findings suggest that this specific type of protein modification plays a significant role in regulating autophagy across various stages. This modification may facilitate the rapid initiation or termination of autophagy in response to diverse cellular demands. Furthermore, our preliminary findings suggest that reduced ATG2A S-palmitoylation also plays a role in the repair of lysosomal damage, thereby underscore the involvement of mTOR signaling in modulating the dynamics of ATG2A S-palmitoylation, including its interaction with ZDHHC11.

## Methods

### Reagents and tools table

| Reagent/Resource | Reference or Source | Identifier or Catalog Number |
|---|---|---|
| **Experimental models** | | |
| HEK293T cells | ATCC | CRL-3216 |
| Hela cells | ATCC | CCL-2 |
| NRK cells | Prof. Li Yu | N/A |
| NRK ATG2A/2B-DKO cells | Prof. Li Yu | N/A |
| HEK293T APT1-KO cells | Prof. Eryan Kong | N/A |
| Escherichia coli BL21(DE3) | Transgen Biotech | CD601 |
| **Recombinant DNA** | | |
| pCMV-HA-ATG2A | This paper | N/A |

| Reagent/Resource | Reference or Source | Identifier or Catalog Number |
| --- | --- | --- |
| pcDNA3.1-Flag-ATG2A | This paper | N/A |
| pcDNA3.1-Flag-ATG9A | This paper | N/A |
| pCMV-HA-ATG2B | This paper | N/A |
| pCMV-HA-ZDHHC11 | This paper | N/A |
| pCMV-HA-ZDHHC6 | This paper | N/A |
| pCMV-HA-APT1 | This paper | N/A |
| pCMV-HA-APT2 | This paper | N/A |
| pEGFP-FIP200 | Prof. Hong Zhang | N/A |
| pcDNA3.1-Flag-ZIKV Envelope Protein | Prof. Geng Li | N/A |
| pCMV-HaloTag-LC3B | MIAOLING BIOLOGY | P69263 |
| pGEX-GST-APT1/2 | This paper | N/A |
| pmCherry-ZDHHC11 | This paper | N/A |
| pmCherry-Sec61β | This paper | N/A |
| pEGFP-ATG2A | This paper | N/A |
| pEGFP-ATG2A-ER | This paper | N/A |
| pCMV-Myc-VMP1 | This paper | N/A |
| pCMV-Myc-WIPI4 | This paper | N/A |
| **Antibodies** | | |
| Rabbit anti-ATG2A | Proteintech | 23226-1-AP |
| Mouse anti-Actin | Proteintech | 66009-1-Ig |
| Rabbit anti-p62 | Proteintech | 18420-1-AP |
| Rabbit anti-LDHA | Proteintech | 19987-1-AP |
| Rabbit anti-APT1 | ABclonal Technology | A4419 |
| Rabbit anti-APT2 | ABclonal Technology | A15792 |
| Rabbit anti-ATG5 | ABclonal Technology | A0203 |
| Rabbit/Mouse anti-HA | Medical & Biological Laboratories | 561-5/M180-3 |
| Rabbit/Mouse anti-Myc | Medical & Biological Laboratories | 562-5/M192-3 |
| Rabbit/Mouse anti-Flag | Medical & Biological Laboratories | PM020/M185-3L |
| Rabbit anti-GFP | Medical & Biological Laboratories | 598 |
| Rabbit anti-ZDHHC11 | GeneTex | GTX106800 |
| Mouse anti-calnexin | BD biosciences | 610523 |
| Anti-mouse IRDye680RD | LI-COR Biosciences | 926-68072 |
| Anti-rabbit IRDye800CW | LI-COR Biosciences | 926-32213 |
| Anti-mouse Alexa Fluor 405-tagged secondary antibody | Molecular Probes | A31553 |
| **Oligonucleotides and other sequence-based reagents** | | |
| Oligonucleotides | This study | Tables EV2, 3 |
| **Chemicals, Enzymes and other reagents** | | |
| Anti-Flag magnetic beads | Bimake | B26102 |
| Anti-HA magnetic beads | Bimake | B26202 |
| Anti-Myc magnetic beads | Bimake | B26302 |

| Reagent/Resource | Reference or Source | Identifier or Catalog Number |
| --- | --- | --- |
| Protein G PLUS-Agarose | Santa Cruz Biotechnology | sc-2002 |
| Protein A Agarose | Santa Cruz Biotechnology | sc-2001 |
| GFP-TRAP magnetic beads | Chromotex | gtma |
| Glutathione-sepharose 4B beads | GE Healthcare Life Sciences | 17-0756-01 |
| Lipofectamine 2000 | Invitrogen | 11668019 |
| Lipofectamine 3000 | Invitrogen | L3000015 |
| TRIzol | Invitrogen | 15596018CN |
| Fluoromount-G | Southern Biotech | 0100-01 |
| DAPI Fluoromount-G | Southern Biotech | 0100-20 |
| Puromycin | Sangon Biotech | E607054 |
| Tris-(2-carboxyethyl)-phosphine hydrochloride | Sangon Biotech | A600974 |
| 4% paraformaldehyde | Sangon Biotech | E672002 |
| phosphatase inhibitor | Sangon Biotech | C500017 |
| Triton X-100 | Sangon Biotech | A110694 |
| protease inhibitor cocktail | Roche | 04693132001 |
| EZ-Link™ Biotin-BMCC | Thermo Scientific | 21900 |
| TAMRA-azide | Thermo Scientific | T10182 |
| LLOMe | MedChemExpress | HY-129905A |
| ML348 | MedChemExpress | HY-100736 |
| ML349 | MedChemExpress | HY-100737 |
| Chloroquine (CQ) | MedChemExpress | HY-17589A |
| SYBR Green qPCR Master Mix (without ROX) | MedChemExpress | HY-K0523 |
| ReverTra Ace qPCR RT Master Mix | TOYOBO | FSQ-201 |
| 2-Bromohexadecanoic acid (2-BP) | Sigma-Aldrich | 21604 |
| Hydroxylamine solution | Sigma-Aldrich | 467804 |
| N-Ethylmaleimide | Sigma-Aldrich | 04260 |
| Palmostatin B | Sigma-Aldrich | 178501 |
| Torin1 | Selleck | S2827 |
| Streptavidin-HRP | Beyotime | A0305 |
| 17-Octadecynoic acid | Cayman | 90270 |
| Lipi-Blue | DOJINDO | LD01 |
| HaloTag® TMR Ligand | Promega | G8252 |
| EBSS | Gibco | 24010043 |
| PBS | Sangon Biotech | E607008 |
| **Software** | | |
| GraphPad Prism | https://www.graphpad.com | N/A |
| ImageJ | https://imagej.net/ij/ | N/A |
| Image Studio Software | https://www.licor.com/bio/support | N/A |

| Reagent/Resource | Reference or Source | Identifier or Catalog Number |
|---|---|---|
| ZEN | Carl Zeiss Microscopy GmbH | N/A |
| CFX Maestro | Bio-Rad | N/A |

## Plasmid constructs

All the plasmids in this study were constructed by using standard molecular cloning techniques. Site-directed mutagenesis was performed using QuikChange II XL following the manufacturer's instructions. The plasmids utilized in this study are detailed in the Reagents and Tools table.

## Cell culture, transfection, and treatment

HEK293T, HeLa, and NRK cells were grown in Dulbecco's modified Eagle medium (DMEM) supplemented with 10% fetal bovine serum (FBS) in a humidified 37 °C incubator with a 5% CO$_2$ atmosphere. Cell lines were recently authenticated by STR profiling and tested for mycoplasma contamination. Cell lines stably expressing the corresponding proteins were established by infecting the cells with retrovirus containing the respective cDNA for 72 h (h), followed by selection using puromycin. For transient transfection of DNA, Lipofectamine 2000 or Lipofectamine 3000 was utilized according to the manufacturer's instructions.

Unless otherwise specified, the cells were treated as follows: For EBSS treatment, cells were cultured in EBSS (Gibco, 24010043) for 6 h after being washed three times with PBS (Sangon Biotech, E607008). For glucose deprivation, cells were maintained in glucose-free DMEM supplemented with 10% FBS for 6 h following 3 washes with PBS. For serum starvation, cells were cultured in DMEM without FBS for 24 h after being washed 3 times with PBS. 2-BP (100 μM) was added to the culture medium and incubated for 24 h. Torin1 (250 nM) was introduced into the culture medium for a duration of 4 h. ML348 (5 μM) was administrated to the culture medium and allowed to act for 24 h. ML349 (20 μM) was added to the culture medium for 36 h. Chloroquine (20 μM) was added to the culture medium and incubated for a period of 16 h.

## RNAi

For RNA interference, siRNA duplexes were synthesized in GenePharma. When the cells reached 70–80% confluence, the siRNA were transfected using Lipofectamine 2000. The culture medium was refreshed 6 h post-transfection. A second transfection was conducted 24 h after the initial transfection. During this second transfection, additional plasmids requiring co-transfection were introduced alongside the siRNA using Lipofectamine 2000, and the cells were harvested 48 h later. The sequences of the siRNAs utilized in this study are listed in Table EV2.

## Western blot and immunoprecipitation

For the Western blot analysis, cells were washed twice with cold PBS and lysed with Laemmeli buffer (60 mM Tris-HCl, 10% glycerol, 2% SDS and 1 mM EDTA, pH 6.8) supplemented with protease and phosphatase inhibitors. Following ultrasonic disruption of the cells, 4 × loading buffer was added to the lysates, which were subsequently boiled in a water bath for 10 min. The proteins within the lysates were separated on SDS polyacrylamide gels and transferred to PVDF membranes (Millipore). After blocking with 5% (w/v) bovine serum albumin (BSA) in TBST, the membrane was incubated overnight at 4 °C with the corresponding primary antibodies followed by incubation at room temperature for 1 h with secondary antibodies. An Odyssey® infrared imaging system (Li-Cor Biosciences) was employed to scan the membranes, while Image J was utilized for quantification of specific protein bands.

For immunoprecipitation, cells were harvested and lysed in NP-40 lysis buffer (50 mM HEPES, 150 mM NaCl, 10% glycerol, 1% NP-40, 1 mM EDTA, 1 mM EGTA, pH 7.4) supplemented with protease and phosphatase inhibitor cocktail. Following ultrasonic disruption of the cells, the cell lysates were centrifuged at 15,000 rpm for 20 min at 4 °C. A small aliquot of the supernatant was taken and mixed with 4 × loading buffer before being boiled in a water bath for 10 min to serve as input control. The remaining supernatant was incubated overnight at 4 °C with specific antibodies. Subsequently, protein A or G agarose that had been pre-cleaned twice with NP-40 lysis buffer was added to the mixture and allowed to incubate for an additional 4 h at 4 °C. After incubation, the immunoprecipitates were washed 4 times using NP-40 lysis buffer and analyzed by Western blotting.

## Immunostaining and cell imaging

For immunostaining, cells growing on coverslips were fixed with 4% paraformaldehyde at room temperature for 15 min. Subsequently, the cells were permeabilized using 0.1% Triton X-100 prepared in PBS for 10 min. To prevent non-specific antibody adsorption, the cells were blocked in PBS containing 10% FBS at room temperature for 1 h before being incubated overnight at 4 °C with the appropriate primary antibodies Following 3 washes with PBS, the cells were incubated at room temperature for 1 h with secondary antibodies, followed by 3 additional washes with PBS. Finally, the samples were mounted using Fluoromount-G or DAPI Fluoromount-G. Images were captured utilizing Airyscan high-resolution mode on the LSM 880 confocal microscope system (Carl Zeiss) and analyzed using ZEN blue software (Carl Zeiss).

## Protein expression and purification

To purify the ATG2A and ATG2A-3CS proteins, Flag-ATG2A and Flag-ATG2A-3CS plasmids were transiently transfected into HEK293S GnTI- cells. Cells were harvested and washed twice with cold PBS before being resuspended in lysis buffer (50 mM Tris-HCl pH 7.4; 200 mM NaCl; 2 mM MgCl$_2$; 10% glycerol; 1% Triton X-100) supplemented with protease inhibitors. The cell suspension was subjected to ultrasonic disruption, followed by centrifugation at 15,000 rpm for 1 h at 4 °C. The supernatant was then incubated for 3 h with anti-Flag affinity beads that had been pre-cleaned twice using lysis buffer. After incubation, the beads were washed 4 times with wash buffer (50 mM HEPES pH 8.0; 200 mM NaCl; 2 mM MgCl$_2$; 1 mM TCEP). To elute ATG2A proteins from anti-Flag beads, wash buffer containing 200 μg/mL Flag peptide was utilized.

GST-APT1/2 were expressed in Escherichia coli BL21 strain through induction with 0.1 mM IPTG for 4 h at 37 °C. Subsequently, the recombinant proteins were subjected to purification utilizing glutathione-sepharose 4B affinity beads. The elution of the proteins was achieved by incubating with glutathione at 4 °C for 4 h.

### In vitro pull-down assay

For GST pull-down assays, purified GST-APT1/2 were incubated with purified Flag-ATG2A proteins overnight at 4 °C. GSH agarose beads were then added to the mixture, followed by further incubation for 4 h at 4 °C. The beads were washed and analyzed using Western blot.

For HA-ZDHHC11 pull-down assays, HEK293T cells transfected with plasmids expressing HA-ZDHHC11 were harvested and lysed. Pre-cleaned anti-HA beads were used to immunoprecipitate HA-ZDHHC11 at 4 °C for 6 h. The beads were washed three times with high-salt IP lysis buffer and then purified Flag-ATG2A proteins were incubated with the beads for 6 h at 4 °C. After incubation, the beads were washed and subjected to Western blot analysis.

### Subcellular fractionations

ER separation was performed as previously described with minor modification (Wong and Adeli, 2009). Following a 48-h transfection of the corresponding plasmid into ATG2A/2B-DKO NRK cells, the cells were harvested and washed with cold PBS. Subsequently, the cells were resuspended in Buffer A (10 mM HEPES, pH 7.2, 0.25 M sucrose, 2 mM EDTA) containing protease inhibitors and homogenized using a Dounce Tissue Grinders. The homogenate was then centrifuged at $8500 \times g$ for 10 min at 4 °C to remove nuclei and mitochondria. The post-nuclear supernatant was collected and centrifugated at $100,000 \times g$ for 3 h at 4 °C. Acetone was added to the supernatant to precipitate proteins, which were subsequently resuspended in loading buffer as cytosolic proteins. The isolation of ER components was involved the overlaying the microsomal pellet with a discontinuous sucrose gradient. Specifically, the pellet was resuspended in 3 mL of 1.22 M sucrose solution in an ultracentrifuge tube. Then, careful layering of 2.6 mL each of 1.15 M, 0.86 M, and 0.25 M sucrose solutions was performed on the microsomal suspension, and this step gradient sucrose solution was centrifugated at $82,000 \times g$ for 3 h at 4 °C. Fractions were collected by unloading 22 aliquots of 500 μL from the top of the tube. Equal volumes of each fraction were prepared for SDS-PAGE. Calnexin immunoblotting was performed to identify which fractions predominantly contained ER component; these identified fractions were utilized as ER fractions for subsequent investigations.

### Purification of GFP-LC3-positive membranes

GFP-LC3-positive membranes was purified as previously described (Gao et al, 2010). Cells were washed with cold PBS and resuspended in Buffer I (0.25 M sucrose, 1 mM EDTA, 20 mM HEPES, pH 7.4) containing a protease inhibitor cocktail. A 22-gauge needle was used to disrupt cells by passing it through the solution 10–15 times. The lysates were then centrifuged at 800 g for 10 min at 4 °C. The post-nuclear supernatant underwent further centrifugation at $10,000 \times g$ for 20 min. Buffer II (PBS, pH 7.4, 0.1% bovine serum albumin, 2 mM EDTA) was utilized to wash the pellets twice for removing residual cytosolic GFP-LC3B present in the pellets. Subsequently, the pellets were suspended in buffer III (2 mM EDTA, 3% BSA in PBS) and incubated with GFP-TRAP magnetic beads for 1 h. Following incubation, the magnetic beads were washed 4 times with cold PBS. The immunoprecipitated samples were subsequently analyzed using Western blotting.

### RNA extraction and quantitative PCR

Total RNA was extracted from cells using TRIzol. The isolated RNA was the reverse transcribed into complementary DNA (cDNA) utilizing the ReverTra Ace qPCR RT Master Mix. Quantitative PCR was performed with SYBR Green qPCR Master Mix (without ROX) on a Real-Time PCR System (Bio-Rad). Each individual sample was normalized against the housekeeping gene Actin. Relative expression changes of each gene were calculated employing the $2^{-\Delta\Delta Ct}$ method. Detailed information regarding the primers used for quantitative PCR is provided in Table EV3.

### Acyl-biotin exchange (ABE) assay

The ABE assay was conducted with minor modifications as previously described (Lu et al, 2019). Briefly, N-ethylmaleimide (NEM) was dissolved in 100% ethanol and added to the lysis buffer (50 mM Tris-HCl pH 7.5, 150 mM NaCl, 1 mM MgCl$_2$, 1% NP-40, 10% glycerol) to achieve a final concentration of 50 mM for blocking free thiol groups. Cells were washed with cold PBS and harvested using the lysis buffer. After ultrasonic disruption, cells were centrifuged at 15,000 rpm for 20 min at 4 °C. A small aliquot of supernatant was mixed with loading buffer and boiled for input control. For exogenously expressed proteins, remaining supernatants were incubated overnight at 4 °C with magnetic beads conjugated to anti-HA or anti-Flag antibodies. For endogenous proteins, specific antibodies were added to the supernatant and incubated overnight at 4 °C before adding protein A or G agarose beads for an additional incubation of 4 h. Following incubation, magnetic or agarose beads were washed with lysis buffer pH 7.5 and washes with lysis buffer adjusted to pH 7.2. Beads from each sample were divided into two Eppendorf tubes, one set was incubated with freshly prepared HAM-containing lysis buffer at pH 7.2 (50 mM Tris-HCl, 150 mM NaCl, 1 mM MgCl$_2$, 1% NP-40, 10% glycerol, 1 M HAM and protease inhibitors), while the other received only the lysis buffer without HAM as a control for blocked free thiols; both sets were kept at room temperature for 1 h. After incubation, all beads underwent washes with lysis buffer pH 7.2 followed by washes using lysis buffer pH 6.2. The beads were then treated with Biotin-BMCC (5 μM) in lysis buffer pH 6.2 on ice for 2 h before being washed sequentially with buffers at both pH 6.2 and pH 7.5. The immunoprecipitates were analyzed by Western blot using specific antibodies alongside streptavidin-HRP.

### 17-ODYA metabolic labeling and click chemistry assay

The 17-ODYA labeling and click chemistry assay for ATG2A was conducted as previously described (Lu et al, 2019). Briefly, Flag-ATG2A-expressing cells were incubated overnight in DMEM with

50 μM 17-ODYA or palmitic acid (PA). After washing with cold D-PBS, the cells were resuspended in D-PBS containing protease inhibitors, Palmostatin B (1 μM), and NEM (50 mM). Cells underwent ultrasonic disruption in an ice bath, followed by centrifuging at 15000 rpm for 20 min at 4 °C. The supernatant was incubated with anti-Flag agarose beads for 3 h and washed 3 times with D-PBS. Subsequently, Cu(I)-assisted click reactions were conducted using D-PBS supplemented with TAMRA-azide (1 μM), TCEP (1 mM), CuSO$_4$ (1 mM) and TBTA (0.1 mM) to incubate the beads for 1 h. After incubation, the beads were washed 8 times with D-PBS and analyzed via fluorescence microscopy. (TCEP stock solution: 50 mM TCEP in H$_2$O adjusted to pH 7.0; CuSO$_4$ stock solution: prepared fresh as 100 mM solution in D-PBS; TBTA stock solution: prepared as a 5 mM solution in an 80% tert-butanol/20% DMSO mixture).

### HaloTag-LC3B processing assay and in-gel fluorescence imaging

The HaloTag-LC3B processing assay for detecting autophagy flux was performed as previously described (Yim et al, 2022). Briefly, cells expressing Halo-LC3B were incubated with 100 nM tetra-methylrhodamine (TMR)-conjugated ligands in complete media at 37 °C for 20 min. The cells were then washed twice with PBS and subsequently incubated in EBSS at 37 °C for 4 h. After incubation, the cells were lysed and the lysates were analyzed by SDS-PAGE. The gel was visualized using the Amersham ImageQuant 800 (cytiva).

### In vitro lipid transfer assay

In vitro lipid transfer assay was performed as previously reported (van Vliet et al, 2022). Briefly, two type of liposomes with different components were prepared: donor liposomes containing DOPC (71%), DOPE (25%), NBD-PE (2%), and Rh-PE (2%); and acceptor liposomes with DOPC (75%) and DOPS (25%). Both liposome types were extruded through a 100 nm filter. Donor and receptor liposomes were mixed to achieve a final lipid concentration of 25 μM each. Purified ATG2A or ATGA-3CS was added to the liposome mixture at a final concentration of 100 nM. Samples were excited at 452 nm and NBD fluorescence intensity was measured at 538 nm.

### HPLC-MS/MS

To identify interacting proteins of ATG2A, HEK293T cells stably expressing HA-vec or HA-ATG2A were lysed, and an immunoprecipitation assay was performed using anti-HA affinity magnetic beads. Each group included three biological replicates. The subsequent steps are as follows: 1. On-beads Digestion: Beads were firstly washed with 200 μL of 100 mM Tris-Cl (pH 8.5), then treated with 50 μL of 8 M urea in 100 mM Tris-Cl (pH 8.5). Reducing agent TCEP (5 mM) and alkylating agent IAA (10 mM) were added, followed by sonication and incubation at room temperature in the dark for 30 min each. The protein mixture was diluted fourfold with 100 mM Tris-Cl and digested with Trypsin at a ratio of 1:50 (w/w). Digestion was halted using 5% Formic Acid (FA), and the peptide mixture was desalted via MonoSpin™ C18 column (GL Science). The desalted mixture was dried using a SpeedVac and resuspended in

0.1% FA for MS analysis. 2. HPLC-tandem MS (MS/MS) Analysis of Peptides: The peptide mixture underwent analysis on a home-made analytical column (30 cm long, pulled-tip, 75 μm ID packed with ReproSil-Pur C18-AQ 1.9 μm resin, Dr. Maisch GmbH) connected to an Easy-nLC 1200 nano HPLC (Thermo Scientific) for mass spectrometry analysis at a column temperature of 55 °C. The mobile phase consisted of buffer A: water with 0.1% formic acid; buffer B: 0.1% formic acid in 80% acetonitrile; elution gradient varied from B concentrations starting at low percentages up to full concentration over specified time intervals. 3. Mass Spectrometry: Data-dependent MS/MS analysis was performed using a Q Exactive Orbitrap mass spectrometer (Thermo Scientific). Peptides eluted from the LC column were electrosprayed into the mass spectrometer at a distal spray voltage of 2.2-kV. A full-scan MS spectrum (*m/z* 300–1800) was acquired, followed by top 20 MS/MS events, generated from the most intense ions selected at 28% normalized collision energy. Full scan resolution was set to 70,000 with an automated gain control (AGC) target of 3e6, while MS/MS scan resolution was set to 17,500 with an isolation window of 1.8 *m/z* and AGC target of 1e5. Both MS and MS/MS scans had one microscan each, with maximum ion injection times of 50 ms and 100 ms, respectively. Dynamic exclusion settings included charge exclusion for charges of +1 and >+8; isotopes exclude; and an exclusion duration, of 15 s. The Xcalibur data system (Thermo Scientific) controlled the MS scan functions and LC solvent gradients. 4. Data Analysis: Acquired MS/MS data were analyzed against a UniProtKB Homo sapiens database using Maxquant V1.6.10.43 with default settings. Precursor mass tolerance and fragment mass tolerance were set to ±20 ppm, while main search peptides tolerance was adjusted to 4.5 ppm based on instrument characteristics in this study.

### Statistical analysis

All the statistical data are presented as mean ± SEM. The statistical significance of differences was determined using Student's *t*-test. $P < 0.05$ was considered statistically significant.

## Data availability

This study includes no data deposited in external repositories.

The source data of this paper are collected in the following database record: biostudies:S-SCDT-10_1038-S44318-025-00410-7.

## Peer review information

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

## Acknowledgements

We thank Dr. Hong Zhang (Institute of Biophysics, Chinese Academy of Sciences, China) and Dr. Geng Li (Guangzhou University of Chinese Medicine, China) for sharing their plasmids; Dr. Li Yu (Tsinghua University, China) and Dr. Eryan Kong (Xinxiang Medical University, China) for providing cell lines; Wei Yin and Guifeng Xiao from the Core Facilities of Zhejiang University School of Medicine for technical support; and Yue Yin from the Mass Spectrometry System at the National Facility for Protein Science (NFPS), Shanghai Advanced Research Institute, Chinese Academy of Science, China for MS technical support. This study was supported by the National Key Research and Development Program of China (2021YFA1300303), the National Natural Science Foundation of China (32230023, 92057203, 31790402, and 91754000), the Fundamental Research Funds for the Central Universities (2020XZZX002-16), and the Innovative Institute of Basic Medical Sciences of Zhejiang University.

## Author contributions

**Wenhui Zheng**: Resources; Data curation; Software; Formal analysis; Investigation; Methodology; Writing—original draft; Writing—review and editing. **Maomao Pu**: Resources; Formal analysis; Investigation; Methodology; Writing—review and editing. **Sai Zeng**: Resources; Formal analysis; Investigation. **Hongtao Zhang**: Resources; Formal analysis; Methodology. **Qian Wang**: Resources; Formal analysis; Investigation. **Tao Chen**: Investigation; Methodology. **Tianhua Zhou**: Resources; Formal analysis. **Chunmei Chang**: Formal analysis; Investigation. **Dante Neculai**: Resources; Methodology. **Wei Liu**: Conceptualization; Supervision; Funding acquisition; Methodology; Writing—original draft; Project administration; Writing—review and editing.

Source data underlying figure panels in this paper may have individual authorship assigned. Where available, figure panel/source data authorship is listed in the following database record: biostudies:S-SCDT-10_1038-S44318-025-00410-7.

## Disclosure and competing interests statement

The authors declare no competing interests.

# Expanded View Figures

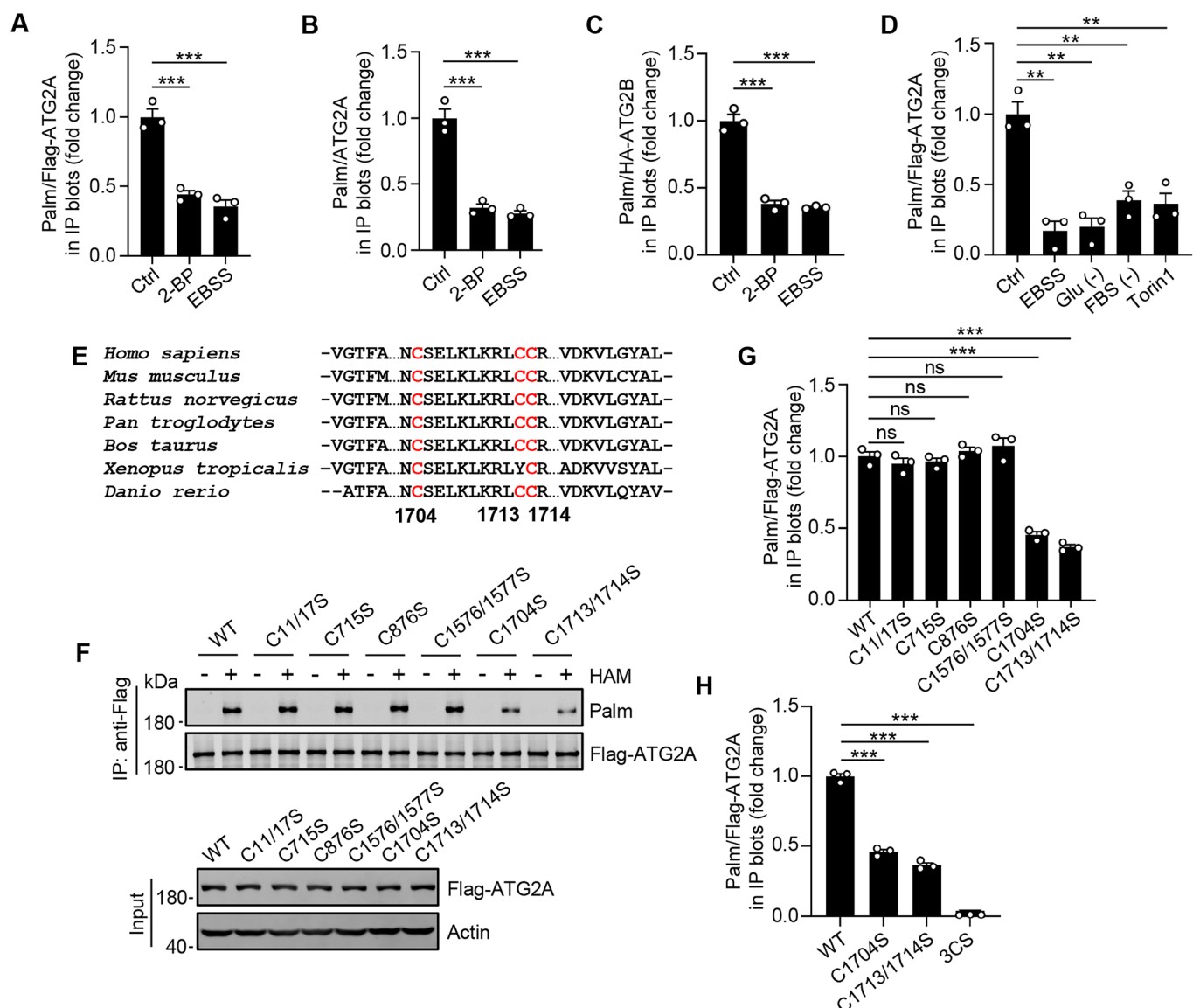

**Figure EV1. Depalmitoylation of ATG2A in starved cells.**

(A–C) Statistical analysis of S-palmitoylation levels for Flag-ATG2A (A), endogenous ATG2A (B) and HA-ATG2B (C) in Fig. 1A–C. Student's *t*-test was used to calculate *P* values. Exact *P* values for (A) from left to right: $P = 0.00096$; $P = 0.00097$. Exact *P* values for (B) from left to right: $P = 0.00084$; $P = 0.00058$. Exact *P* values for (C) from left to right: $P = 0.00034$; $P = 0.00019$. (D) Statistical analysis of S-palmitoylation levels for Flag-ATG2A in Fig. 1E. Student's *t*-test was used to calculate *P* values. Exact *P* values from left to right: $P = 0.00149$; $P = 0.00164$; $P = 0.00449$; $P = 0.00499$. (E) Conservation analysis of S-palmitoylation sites in ATG2A across species. (F) S-palmitoylation levels of Flag-ATG2A mutants measured by ABE assay in HEK293T cells. (G) Statistical analysis of (F). Student's *t*-test was used to calculate *P* values. Exact *P* values from left to right: $P = 0.38347$; $P = 0.44816$; $P = 0.40234$; $P = 0.30211$; $P = 0.00017$; $P = 7.48E{-}05$. (H) Statistical analysis of the S-palmitoylation level for Flag-ATG2A mutants in Fig. 1G. Student's *t*-test was used to calculate *P* values. Exact *P* values from left to right: $P = 3.80E{-}05$; $P = 2.24E{-}05$; $P = 1.41E{-}06$. Data information: All statistical data are presented as mean ± SEM of three independent experiments. ns, not significant; **$P < 0.01$; ***$P < 0.001$ (Student's *t*-test). Source data are available online for this figure.

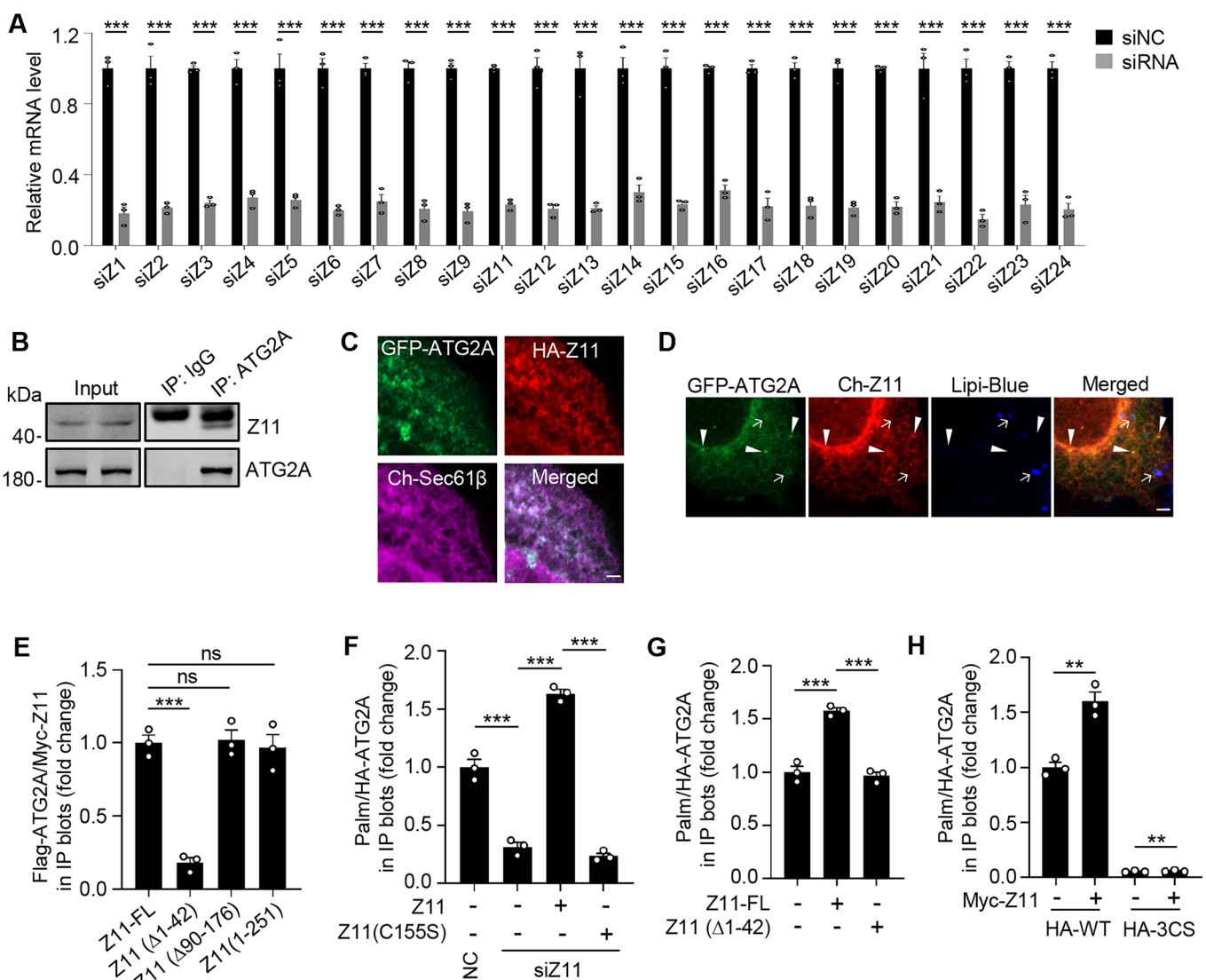

**Figure EV2. ATG2A is S-palmitoylated by ZDHHC11.**

(A) qPCR analysis of ZDHHC gene RNAi efficiency in HEK293T cells. qPCR primers for ZDHHC11 are designed to target ZDHHC11 and ZDHHC11B. Student's *t*-test was used to calculate *P* values. Exact *P* values from left to right: $P = 0.00018$; $P = 0.00040$; $P = 4.62E−06$; $P = 0.00026$; $P = 0.00096$; $P = 0.00017$; $P = 0.00012$; $P = 0.00012$; $P = 6.00E−05$; $P = 3.25E−06$; $P = 0.00027$; $P = 0.00064$; $P = 0.00069$; $P = 0.00025$; $P = 2.83E−05$; $P = 0.00011$; $P = 8.83E−05$; $P = 8.22E−05$; $P = 9.73E −06$; $P = 0.00122$; $P = 0.00015$; $P = 0.00031$; $P = 0.00011$. (B) Co-immunoprecipitation of endogenous ZDHHC11 and ATG2A in HEK293T cells. (C) Representative images showing the localization of GFP-ATG2A, HA-ZDHHC11 and Cherry-Sec61β. GFP-ATG2A, HA-ZDHHC11 and Cherry-Sec61β were transfected in HeLa cells cultured in complete media and the cells were imaged 48 h post-transfection. Scale bars, 2 μm. (D) Representative images showing the localization of GFP-ATG2A, HA-ZDHHC11 and lipid droplets. GFP-ATG2A and Cherry-ZDHHC11 were transfected in HeLa cells cultured in complete media and the cells were stained with Lipi-Blue and imaged 48 h post-transfection. Arrows point to the colocalization sites of ATG2A with LDs. Arrowheads indicate the sites of colocalization between ATG2A and ZDHHC11. Scale bar, 2 μm. (E) Statistical analysis of ratio of Flag-ATG2A to Myc-ZDHHC11 in IP blots for Fig. 2E. Student's *t*-test was used to calculate *P* values. Exact *P* values from left to right: $P = 0.00021$; $P = 0.82154$; $P = 0.77098$. (F–H) Statistical analysis of the S-palmitoylation level for HA-ATG2A in Fig. 2F–H. Student's *t*-test was used to calculate *P* values. Exact *P* values for (F) from left to right: $P = 0.00084$; $P = 1.47E−05$; $P = 5.63E−06$. Exact *P* values for (G) from left to right: $P = 0.00069$; $P = 0.00014$. Exact *P* values for (H) from left to right: $P = 0.00332$; $P = 0.50351$. Data information: All statistical data are presented as mean ± SEM of three independent experiments. ns, not significant; **$P < 0.01$; ***$P < 0.001$ (Student's *t*-test). Source data are available online for this figure.

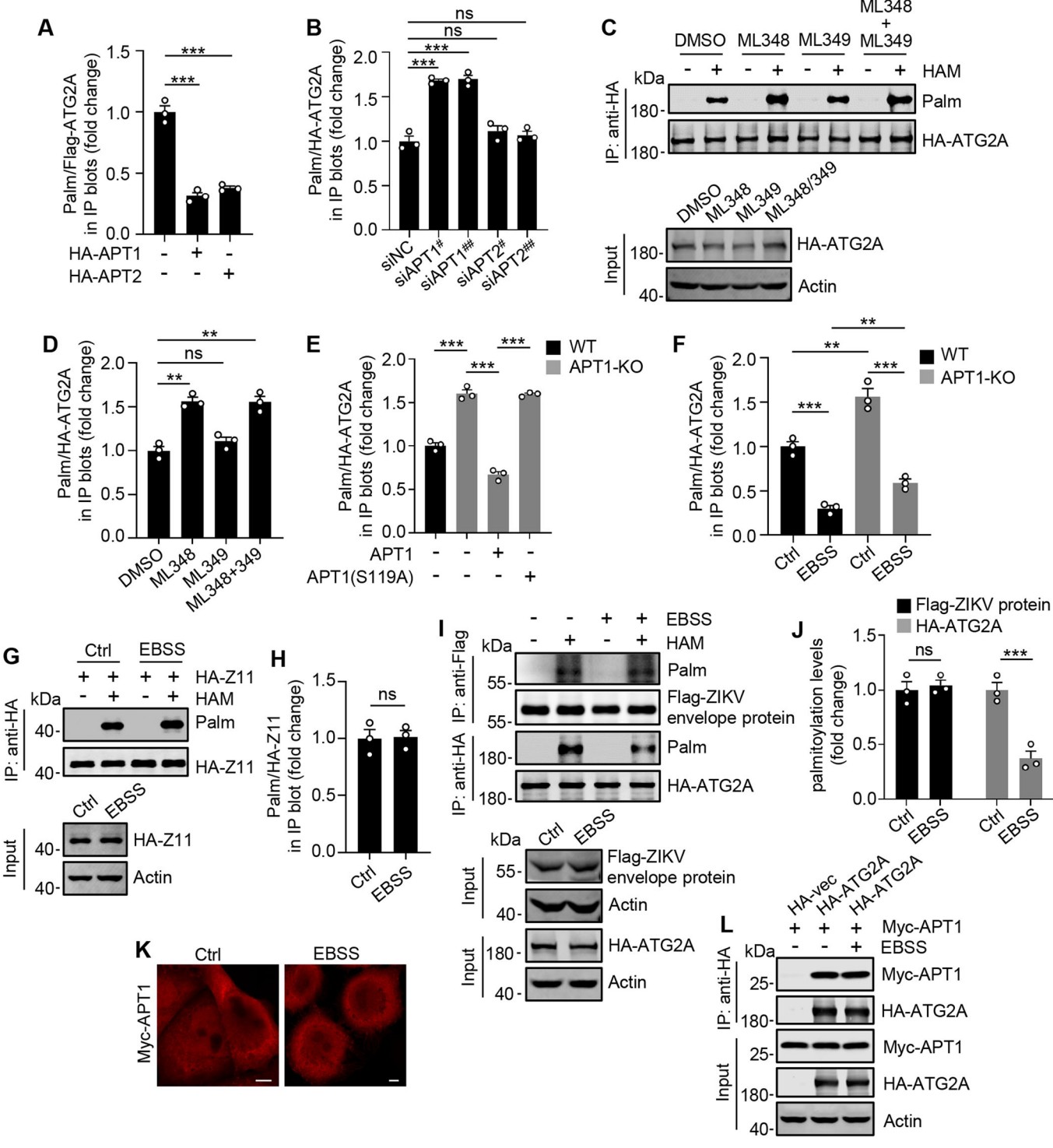

◀ **Figure EV3.  APT1 is a depalmitoylase of ATG2A.**

(A, B) Statistical analysis of the S-palmitoylation level for Flag-ATG2A or HA-ATG2A in Fig. 3C, D. Student's *t*-test was used to calculate *P* values. Exact *P* values for (A) from left to right: $P = 0.00032$; $P = 0.00034$. Exact *P* values for (B) from left to right: $P = 0.00037$; $P = 0.00069$; $P = 0.22506$; $P = 0.37955$. (C) S-palmitoylation of HA-ATG2A in HEK293T cells expressing HA-ATG2A. The cells were treated with specific APT inhibitor. (D) Statistical analysis of (C). Student's *t*-test was used to calculate *P* values. Exact *P* values from left to right: $P = 0.00107$; $P = 0.18353$; $P = 0.00233$. (E, F) Statistical analysis of the S-palmitoylation level for HA-ATG2A in Fig. 3E, F. Student's *t*-test was used to calculate *P* values. Exact *P* values for (E) from left to right: $P = 0.00039$; $P = 7.69E{-}05$; $P = 1.80E{-}05$. Exact *P* values for (F) from left to right: $P = 0.00046$; $P = 0.00665$; $P = 0.00576$; $P = 0.00067$. (G) ABE assay of HA-ZDHHC11 S-palmitoylation in HA-ZDHHC11-transfected HEK 293T cells cultured with or without EBSS medium. (H) Statistical analysis of (G). (I) S-palmitoylation of Flag-tagged ZIKV envelope protein and HA-ATG2A in HEK293T cells cultured with or without EBSS medium. (J) Statistical analysis of (I). Student's *t*-test was used to calculate *P* values. Exact *P* values from left to right: $P = 0.66553$; $P = 0.00287$. (K) Representative images showing the localization of Myc-APT1 in Hela cells with or without EBSS treatment. (L) Co-immunoprecipitation of APT1 with ATG2A. HA-ATG2A was immunoprecipitated from HA-ATG2A-expressing HEK293T cells transfected with Myc-APT1 with or without EBSS treatment. The precipitates were analyzed by Western blot using anti-Myc. Data information: All statistical data are presented as mean ± SEM of three independent experiments. ns, not significant; \*\**P* < 0.01; \*\*\**P* < 0.001 (Student's *t*-test). Source data are available online for this figure.

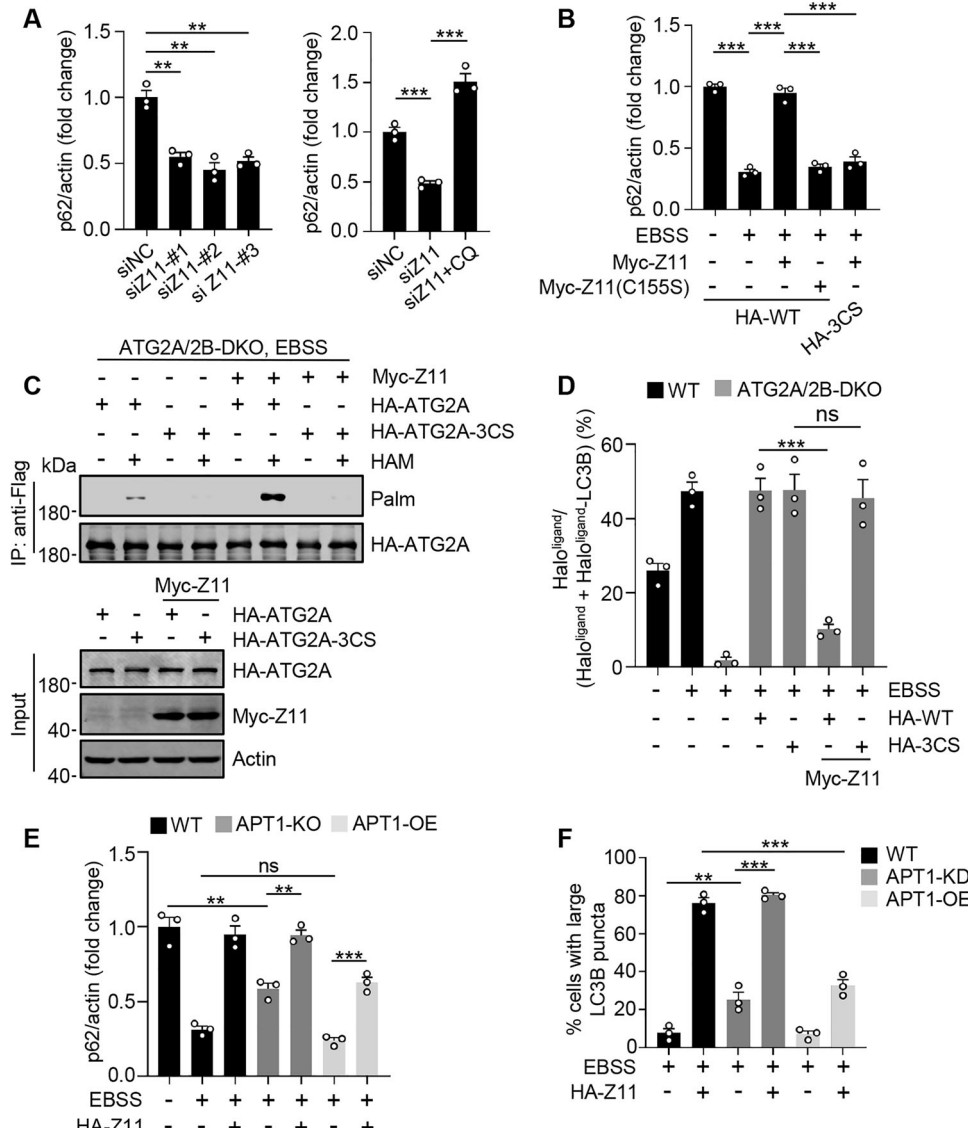

**Figure EV4. Depalmitoylation of ATG2A in starvation-induced autophagy.**

(A, B) Statistical analysis of p62 protein levels in Fig. 4A, C. Student's *t*-test was used to calculate *P* values. Exact *P* values for (A) from left to right: $P = 0.00203$; $P = 0.00196$; $P = 0.00150$; $P = 0.00065$; $P = 0.00026$. Exact *P* values for (B) from left to right $P = 2.23E-05$; $P = 0.00014$; $P = 0.00018$; $P = 0.00050$. (C) S-palmitoylation of HA-ATG2A or HA-ATG2A-3CS in ATG2A/2B-DKO NRK cells stably expressing HA-ATG2A or HA-ATG2A-3CS. The cells were transfected with or without Myc-ZDHHC11 and treated with EBSS. (D) Statistical analysis of Fig. 4H. Student's *t*-test was used to calculate *P* values. Exact *P* values from left to right: $P = 0.00050$; $P = 0.76680$. (E) Statistical analysis of p62 protein levels in Fig. 4I. Student's *t*-test was used to calculate *P* values. Exact *P* values from left to right: $P = 0.00514$; $P = 0.06602$; $P = 0.00209$; $P = 0.00064$. (F) Statistical analysis of the proportion of cells containing large LC3B puncta (diameter > 1.5 μm) in Fig. 4J. Student's *t*-test was used to calculate *P* values. Exact *P* values from left to right: $P = 0.01556$; $P = 0.00016$; $P = 0.00051$. Data information: All statistical data are presented as mean ± SEM of three independent experiments. ns, not significant; **$P < 0.01$; ***$P < 0.001$ (Student's *t*-test). Source data are available online for this figure.

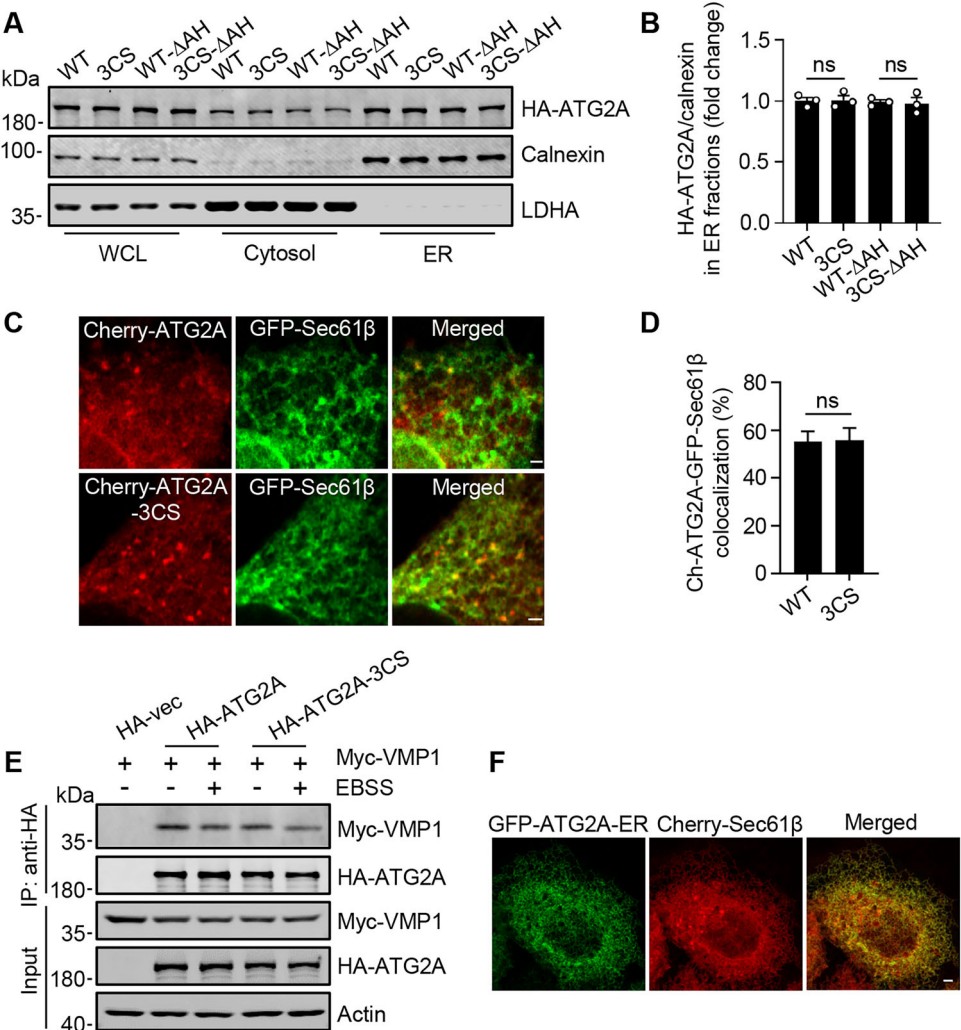

**Figure EV5. S-palmitoylation anchors ATG2A C-terminal to the ER.**

(**A**) Western blot analysis of the distribution of ATG2A and its mutants in subcellular fractions of ATG2A/2B-DKO NRK cells. Cells were transfected with HA-ATG2A, HA-ATG2A-3CS, HA-ATG2A-ΔAH, or HA-ATG2A-3CS-ΔAH. WCL: whole cell lysis. (**B**) Statistical analysis of (**A**). (**C**) Representative images showing the colocalization of Cherry-ATG2A or Cherry-ATG2A-3CS with GFP-Sec61β in HeLa cells. Scale bars, 1 μm. (**D**) Statistical analysis of (**C**). $n = 30$ cells. (**E**) Co-immunoprecipitation of VMP1 with ATG2A or ATG2A-3CS. HA-ATG2A or HA-ATG2A-3CS was immunoprecipitated from HEK293T cells co-transfected with Myc-VMP1. The cells were treated with or without EBSS, and the precipitates were analyzed by Western blot using anti-Myc. (**F**) Representative images showing the colocalization of GFP-ATG2A-ER and Cherry-Sec61β in HeLa cells. Scale bar, 2 μm. Data information: All statistical data are presented as mean ± SEM of three independent experiments unless otherwise specified. ns, not significant (Student's *t*-test). Source data are available online for this figure.

