## [Peer Review File · The EMBO Journal]

S-palmitoylation modulates ATG2-dependent non-vesicular lipid transport during starvation-induced autophagy

Wenhui Zheng, Maomao Pu, Sai Zeng, Hongtao Zhang, Qian Wang, Tao Chen, Tianhua Zhou, Chunmei Chang, Dante Neculai, and Wei Liu

Corresponding author(s): Wei Liu (liuwei666@zju.edu.cn) , Dante Neculai (dneculai@zju.edu.cn), Chunmei Chang (chunmei_chang@fudan.edu.cn)

Review Timeline:

Submission Date:	26th Sep 24
Editorial Decision:	7th Nov 24
Revision Received:	25th Jan 25
Editorial Decision:	17th Feb 25
Revision Received:	22nd Feb 25
Accepted:	2nd Mar 25

Editor: William Teale

Transaction Report:

Dear Prof. Liu,

Thank you again for the submission of your manuscript entitled "Depalmitoylation activates lipid transfer protein ATG2 in starvation-induced autophagy" and for your patience during the review process. We have now received the reports from the referees, which I copy below.

As you can see from their comments, all acknowledge that your work is timely and well presented. That said, all of them point out specific issues that will require your attention before your manuscript can be published in The EMBO Journal.

Based on the overall interest expressed in the reports though, I would like to invite you to address the comments of all referees in a revised version of the manuscript. I should add that it is The EMBO Journal policy to allow only a single major round of revision and that it is therefore important to resolve the main concerns at this stage. I believe the concerns of the referees are reasonable and addressable, but please contact me if you have any questions, need further input on the referee comments or if you anticipate any problems in addressing any of their points. Please, follow the instructions below when preparing your manuscript for resubmission.

I would also like to point out that as a matter of policy, competing manuscripts published during this period will not be taken into consideration in our assessment of the novelty presented by your study ("scooping" protection). We have extended this 'scooping protection policy' beyond the usual 3 month revision timeline to cover the period required for a full revision to address the essential experimental issues. Please contact me if you see a paper with related content published elsewhere to discuss the appropriate course of action.

Again, please contact me at any time during revision if you need any help or have further questions.

Thank you very much again for the opportunity to consider your work for publication. I look forward to your revision.

Best regards,

William

William Teale, Ph.D.
Editor
The EMBO Journal

When submitting your revised manuscript, please carefully review the instructions below and include the following items:

- 1) a .docx formatted version of the manuscript text (including legends for main figures, EV figures and tables). Please make sure that the changes are highlighted to be clearly visible.
- 2) individual production quality figure files as .eps, .tif, .jpg (one file per figure).
- 3) a .docx formatted letter INCLUDING the reviewers' reports and your detailed point-by-point response to their comments. As part of the EMBO Press transparent editorial process, the point-by-point response is part of the Review Process File (RPF), which will be published alongside your paper.
- 4) a complete author checklist, which you can download from our author guidelines ([https://wol-prod-cdn.literatumonline.com/pb-assets/embo-site/Author Checklist%20-%20EMBO%20J-1561436015657.xlsx](https://wol-prod-cdn.literatumonline.com/pb-assets/embo-site/Author%20Checklist%20-%20EMBO%20J-1561436015657.xlsx)). Please insert information in the checklist that is also reflected in the manuscript. The completed author checklist will also be part of the RPF.
- 5) Please note that all corresponding authors are required to supply an ORCID ID for their name upon submission of a revised manuscript.
- 6) We require a 'Data Availability' section after the Materials and Methods. Before submitting your revision, primary datasets produced in this study need to be deposited in an appropriate public database, and the accession numbers and database listed under 'Data Availability'. Please remember to provide a reviewer password if the datasets are not yet public (see <https://www.embopress.org/page/journal/14602075/authorguide#datadeposition>). If no data deposition in external databases is

needed for this paper, please then state in this section: This study includes no data deposited in external repositories. Note that the Data Availability Section is restricted to new primary data that are part of this study.

Note - All links should resolve to a page where the data can be accessed.

8) For data quantification: please specify the name of the statistical test used to generate error bars and P values, the number (n) of independent experiments (specify technical or biological replicates) underlying each data point and the test used to calculate p-values in each figure legend. The figure legends should contain a basic description of n, P and the test applied. Graphs must include a description of the bars and the error bars (s.d., s.e.m.).

9) We would also encourage you to include the source data for figure panels that show essential data. Numerical data can be provided as individual .xls or .csv files (including a tab describing the data). For 'blots' or microscopy, uncropped images should be submitted (using a zip archive or a single pdf per main figure if multiple images need to be supplied for one panel). Additional information on source data and instruction on how to label the files are available at .

10) We replaced Supplementary Information with Expanded View (EV) Figures and Tables that are collapsible/expandable online (see examples in <https://www.embopress.org/doi/10.15252/embj.201695874>). A maximum of 5 EV Figures can be typeset. EV Figures should be cited as 'Figure EV1, Figure EV2" etc. in the text and their respective legends should be included in the main text after the legends of regular figures.

12) Our journal encourages inclusion of *data citations in the reference list* to directly cite datasets that were re-used and obtained from public databases. Data citations in the article text are distinct from normal bibliographical citations and should directly link to the database records from which the data can be accessed. In the main text, data citations are formatted as follows: "Data ref: Smith et al, 2001" or "Data ref: NCBI Sequence Read Archive PRJNA342805, 2017". In the Reference list, data citations must be labeled with "[DATASET]". A data reference must provide the database name, accession number/identifiers and a resolvable link to the landing page from which the data can be accessed at the end of the reference. Further instructions are available at .

13) In order to increase the reproducibility and reach of your work, The EMBO Journal includes a table of reagents that were used in the study. Please provide this along with your revisions.

We realize that it is difficult to revise to a specific deadline. In the interest of protecting the conceptual advance provided by the work, we recommend a revision within 3 months (5th Feb 2025). Please discuss the revision progress ahead of this time with the editor if you require more time to complete the revisions. Use the link below to submit your revision:

Referee #1:

In this manuscript, Zheng and colleagues identified that Atg2 is regulated by palmitoylation/depalmitoylation on its C-terminal three cysteines during autophagy. The authors identify ZDHHC11 and APT1 as playing such roles. Palmitoylation does not interfere with the lipid transfer activity of Atg2; it does not allow it to interact with phagophores and negatively regulate autophagy. When starving cells, Atg2 was found to be depalmitoylated and found with LC3-II; thus, authors claimed that depalmitoylated Atg2 can transfer lipids to the phagophore. The palmitoylation state of Atg2 was clearly shown in all conditions; questions remain about the mechanism of how it is regulated and thus functions in autophagy. Overall, the results were clean and displayed nicely. Their discovery advances our understanding of the molecular consequences of autophagosome biogenesis in autophagy.

Major concerns

1. How does Atg2 get depalmitoylated by APT1 under the starvation condition? Where does APT1 localize under nutrient and starved conditions? If APT1 is a responsive enzyme, authors need to show the palmitoylation state of Atg2 in KD or KO of APT1. The authors used inhibitors; however, they were not additives under EBSS. Does EBSS treatment affect the interaction between ATG2 and APT1 or the colocalization state? The interaction of Atg2 and ZDHHC11 was decreased in EBSS-treated cells, and whether APT1 plays a role or not, authors need to show Atg2-ZDHHC11 interactions in APT1 KD or KO cells.
2. The authors show the interaction between Atg2 and Atg9. Such interaction becomes more robust under starved conditions and HA-3CS Atg2. Thus, the authors concluded in the model that dephosphorylated Atg2 transfers lipids to phagophores through Atg9. Majority of Atg9 exits as Atg9 vesicles even under the starved condition. How do authors know that Atg2 transfers lipids to phagophores and not to Atg9 vesicles in the experimental conditions? If depalmitoylated Atg2 only interacts with Atg9 on phagophores, how does Atg2 distinguish between the two? The authors showed palmitoylation state does not affect the lipid transfer activity of Atg2. The title needs to be more accurate with the data presented in this paper if the authors could show that the depalmitoylation of Atg2 regulated the lipid transfer activity.
3. Please show the palmitoylation state of Atg2 in Atg2A/2B-DKO +Atg2A Wt + Myc-Z11. Why were small LC3 dots found more in these cells compared to the control (Atg2A/2B-DKO + Myc Vec or DKO itself)?

Minor concerns

Figure legends need to describe the experimental conditions for Fig. 3A and Fig. 6H. Are cells starved?

Referee #2:

The manuscript from Zheng et al. addresses the important question of how phospholipid flux across the membrane contact site between ER and phagophore mediated by ATG2, a conserved protein family of rod-like phospholipid transfer proteins, is regulated. They describe a novel depalmitoylation-dependent activation mechanism of ATG2, which is important for autophagy and lysosomal damage repair. In this study the authors report that, under nutrient rich conditions, ATG2A is post-translationally modified at three conserved cysteine residues at its C-terminus by the S-palmitoyltransferase ZDHHC11. This modification tethers the C-terminus of ATG2A to the ER membrane without interfering with the lipid transfer activity of the protein. During starvation, ATG2A is depalmitoylated by the depalmitoylase APT1, which releases the ATG2A C-terminus from the ER membrane and enables the interaction of ATG2A with the phagophore membrane - an important step during autophagy which yields in the formation of autophagosomes.

The manuscript describes data suggesting a new mechanism of how ATG2 binding to the phagophore might be regulated to control autophagosome biogenesis. Overall, the data are convincing and supports the conclusions of the authors. However, there are a few points that need to be addressed:

Major comments:

1. It is required for good scientific practice and rigor that all Western blot analyses with quantitative interpretations are carefully quantified ($n \geq 3$) and statistically analyzed.
2. Figure 2E: The authors argue that deletion of the first 42 residues of Z11 leads to disruption in the interaction between Z11 and ATG2A, because they see reduction in ATG2A band intensity. However, the amounts of immunoprecipitated Myc-Z11 vary between samples making it difficult to link the drop in ATG2A band intensity to impaired interaction between the two proteins or to the different amounts of Z11 proteins precipitated in this set-up. The authors need to (1) show less overexposed WBs and (2) quantify each ratio of ATG2A and the tested Z11 constructs to test for changes in interaction between ATG2A and the Z11 Δ 1-42 mutant. Additionally, to assess the biological outcome independently, the authors should check the localization of ATG2A and Z11 constructs in fluorescence microscopy using SEC61 labeled cells.
3. Fig. 2G: Authors should show the IP blot of Z11 and Z11 Δ 1-42 constructs and quantify the ratio of Palm/Z11 or Z11 Δ 1-42 to quantitatively show the increase in ATG2A S-palmitoylation which they suggest in the main text.
4. Fig. 2H: Authors should show the IP blot for Myc-Z11 and quantify the ratio of Palm/Z11.
5. Fig. 3B: Authors should show the IP blot for HA-APT1/2 and quantify the ratio of Palm/APT1 and APT2 and Palm/ATG2A.
6. Fig. 3C: Authors should show the IP blot for APT1/2 and quantify the ratio of Palm/ATG2A.
7. Fig. 3E: authors should show the IP blots for Myc-APT1 and Myc-APT1 (S119A).
8. The authors use a CoIP approach to study the interaction between ATG2A and ZDHHC11 and APT1/2. However, while this approach is consistent with direct interactions between ATG2A and Z11 or APT1/2, it does not demonstrate it, and it is formally possible that they interact indirectly via additional proteins. Demonstrating direct interactions between ATG2A and Z11 or APT1/2 would require an in vitro binding assay with purified proteins.
9. The authors investigate how the loss or overexpression of Z11 affects the starvation-induced degradation of p62 and the formation of LC3 puncta in starved cells. They further suggest that inhibition of the depalmitoylation of ATG2 replicates loss of ATG2 and results in the accumulation of large LC3 structures in the cytosol of starved cells. However, the authors only examine the effects of Z11 overexpression in these experimental set-ups. Since the authors identified APT1 as the major depalmitoylase for ATG2A, they should investigate how loss and overexpression of APT1 affects the degradation of p62 and the LC3-dot formation in the absence or presence of overexpressed Z11.
10. Fig EV2 A and B: The authors should include a control in which an ATG2 mutant is not anchored at the ER but still retains a connection to the growing phagophore and quantify the localization of ATG2 puncta with the ER in each condition.

Minor comments:

1. Some figures throughout the manuscript are incompletely labeled and insufficiently described in the figure legends. For example, in Fig.1A, B, C authors use HAM (hydroxylamine), without describing what HAM is in the main text and figure legend. In Figure 1D, authors use palmitic acid, however, they do not mention it in the figure legend. Carefully describing all sample treatments in the figure legends throughout the manuscript will be beneficial for the reader.
2. Figure 1A, 1B, 1C, 1E, 1F,1H,1I, 2A, 3D, 3G, 3H,6I: authors should add the Input blots.
3. Fig. 2B: Authors should correctly label the bands for Z11 and Z6 in the shown Western Blot image.
4. Figure 2D: Can the authors specify if the cells were grown in full media and add this information to the figure legend?
5. Fig. 2F: authors should show the IP blot for Z11 constructs.
6. Fig. 4E: a negative control for LC3-dot formation is missing. The authors should include as a control cells grown in nutrient rich conditions.
7. The authors show that ATG2A S-palmitoylation affects the interaction between ATG9 and ATG2A. Is ATG9 required for ATG2A depalmitoylation?

Referee #3:

ATG2A is a lipid transfer protein that bridges ER and phagophore and transfers lipids from ER to phagophore for membrane

elongation to build up an autophagosome. Although the lipid transfer activity of ATG2A has been well characterized, the regulation mechanism of the activity remains largely unknown. In this manuscript, the authors found that ATG2A is palmitoylated at Cys1704, 1713, and 1714 by ER-resident palmitoyltransferase ZDHHC11 and depalmitoylated by depalmitoylase APT1. ZDHHC11 directly bound ATG2A and showed colocalization with ATG2A. Expression of ATG2A with mutations at palmitoylation sites enhanced autophagy activity but the same mutations did not enhance the lipid transfer activity of purified ATG2A. Further analyses showed that palmitoylation of ATG2A made the C-terminal region of ATG2A to bind to ER and thereby inhibit the ER-phagophore bridging. Based on these data, the authors concluded that the lipid transfer activity of ATG2A is negatively regulated by palmitoylation by making the C-terminus of ATG2A to bind to ER and activated by depalmitoylation which enables ATG2A to bridge between ER and phagophore and transfer lipids from ER to phagophore for autophagosome formation. ATG2-mediated lipid transfer is one of the critical steps in autophagosome formation and its regulation by palmitoylation is novel and interesting. The experiments are well designed and the obtained results are clear and convincing. Thus this manuscript will be attractive to the broad readers of EMBO Journal. However, one of the major conclusions is inconsistent with previous reports and need to be resolved.

Major points

- 1) The main conclusion that ATG2A localizes predominantly to ER by S-palmitoylation is not consistent with previous reports showing that ATG2A mainly localizes at lipid droplets and phagophores, but not at ER (PMID 22219374, 29113029). In Fig. 2D, the authors observed that ATG2A formed punctual structures with ZDHHC11; however, in contrast to the ER pattern of ZDHHC11, ATG2A did not show such pattern. The punctual structures seem to be lipid droplets rather than ER. In Fig. EV2B, the authors compared the localization of ATG2A and Sec61beta, an ER-marker, which also showed that ATG2A colocalizes with Sec61beta as puncta but not as ER pattern. Analyze the colocalization of ATG2A, ZDHHC11, lipid droplet marker, and ER marker and clearly indicate whether the colocalization site of ATG2A and ZDHHC11 is ER or lipid droplets. If the site corresponds to lipid droplets, reconsider the model shown in Fig. 6L.
- 2) Recently, a Halo-tag assay (PMID 35938926) was developed to more accurately quantify the autophagic flux compared with conventional methods used in this manuscript. Perform Halo-tag assay to further confirm the effect on the autophagic flux.

Minor points

- 1) As an example of S-palmitoylation in the regulation of autophagy, a recent manuscript (PMID 39362856) should also be cited in the Introduction.
- 2) As to Fig. 1A-C, add a brief explanation of the ABE assay in the main text and of Palm in the legend.
- 3) In Fig. 1D, cells that do not express Flag-ATG2A and have been treated with 17-ODYA should be added as a negative control.
- 4) The authors discussed that "S-palmitoylation only affects the interaction between ATG2A and Atg9, but not with WIPI4. This may be due to the fact that the ATG9-binding domain (aa 1760-1779) on ATG2A is closer to the S-palmitoylation sites than the WIPI4-binding domain (aa 1358-1404)" in page 16. However, based on the model proposed by the authors, this reviewer thinks that S-palmitoylation anchors the C-terminus of Atg2 to the ER, making it unable to interact with the autophagic membrane, and therefore, unable to interact with ATG9, a transmembrane protein that does not localize to the ER but to the autophagic membrane (in contrast, WIPI4 is not a transmembrane protein and thus can move to the ER surface and binds to ER-anchored ATG2A). Reconsider the discussion.

Response to the referees**Referee #1:**

In this manuscript, Zheng and colleagues identified that Atg2 is regulated by palmitoylation/depalmitoylation on its C-terminal three cysteines during autophagy. The authors identify ZDHHC11 and APT1 as playing such roles. Palmitoylation does not interfere with the lipid transfer activity of Atg2; it does not allow it to interact with phagophores and negatively regulate autophagy. When starving cells, Atg2 was found to be depalmitoylated and found with LC3-II; thus, authors claimed that depalmitoylated Atg2 can transfer lipids to the phagophore. The palmitoylation state of Atg2 was clearly shown in all conditions; questions remain about the mechanism of how it is regulated and thus functions in autophagy. Overall, the results were clean and displayed nicely. Their discovery advances our understanding of the molecular consequences of autophagosome biogenesis in autophagy.

Re: We sincerely appreciate your positive evaluation of our study. We have performed additional experiments to further investigate the regulatory mechanism and functions associated with ATG2A depalmitoylation in autophagy.

Major concerns

1. How does Atg2 get depalmitoylated by APT1 under the starvation condition? Where does APT1 localize under nutrient and starved conditions? If APT1 is a responsive enzyme, authors need to show the palmitoylation state of Atg2 in KD or KO of APT1. The authors used inhibitors; however, they were not additives under EBSS. Does EBSS treatment affect the interaction between ATG2 and APT1 or the colocalization state? The interaction of Atg2 and ZDHHC11 was decreased in EBSS-treated cells, and whether APT1 plays a role or not, authors need to show Atg2-ZDHHC11 interactions in APT1 KD or KO cells.

Re: Our results indicate that the observed depalmitoylation of ATG2A (decreased palmitoylation levels) in cells under starvation is primarily attributed to the inhibition of the interaction between ZDHHC11 and ATG2A.

In response to your requests and suggestions, we conducted additional experiments which revealed that APT1 is predominantly localized in the cytoplasm, and EBSS treatment did not alter either the localization of APT1 or its interaction with ATG2A. Furthermore, we examined the palmitoylation status of ATG2A and its interaction with ZDHHC11 in APT1-KO cells. The results demonstrated that EBSS treatment still resulted in a reduction of ATG2A palmitoylation and a comparable interaction between ATG2A and ZDHHC11 in the absence of APT1. These data reinforce our conclusions that while deletion of APT1 can elevate basal palmitoylation levels of ATG2A, the observed decrease in ATG2A palmitoylation during starvation, essential for starvation-induced autophagy, is primarily due to a weakened interaction between ATG2A and ZDHHC11, rather than significant activation of APT1.

These new data have been incorporated as Fig. EV3K, Fig. EV3L, Figs. 3F and EV3F, and Fig. 3G, H respectively in the revised manuscript.

2. The authors show the interaction between Atg2 and Atg9. Such interaction becomes more robust under starved conditions and HA-3CS Atg2. Thus, the authors concluded in the model that dephosphorylated Atg2 transfers lipids to phagophores through Atg9. Majority of Atg9 exits as Atg9 vesicles even under the starved condition. How do authors know that Atg2 transfers lipids to phagophores and not to Atg9 vesicles in the experimental conditions? If depalmitoylated Atg2 only interacts with Atg9 on phagophores, how does Atg2 distinguish between the two? The authors showed palmitoylation state does not affect the lipid transfer activity of Atg2. The title needs to be more accurate with the data presented in this paper if the authors could show that the depalmitoylation of Atg2 regulated the lipid transfer activity.

Re: Accumulating studies utilizing yeast (PMID 32883836) and mammalian cells (PMID 37115958, 37115157) indicate that Atg9 vesicles serve as seeds for the isolation membrane formation. The Atg2-Atg9 interaction is crucial for the growth and elongation of isolation membranes during starvation-induced autophagy by facilitating lipid transfer to the membranes. Currently, it remains

unclear whether Atg2 can distinguish between Atg9 in vesicles and those present in phagophores; no study has yet examined whether Atg2 transfers lipids to Atg9 vesicles prior to their fusion into isolation membranes. This investigation would be particularly challenging due to the dynamic nature of vesicles within cells. Nevertheless, this does not alter the conclusion drawn from our study. Additionally, we have shown that depalmitoylation is necessary for ATG2A to target FIP200 puncta, another marker associated with phagophores. The examination of the interaction between Atg2 and Atg9 was primarily intended to support these observations.

Our results suggest that the depalmitoylation of ATG2A plays a crucial role regulating autophagy by modulating its interaction with phagophores, rather than influencing its intrinsic lipid transfer activity. Thank you for your suggestion, to preventing any potential misunderstandings, we have revised the title of our manuscript.

3. Please show the palmitoylation state of Atg2 in Atg2A/2B-DKO +Atg2A Wt + Myc-Z11. Why were small LC3 dots found more in these cells compared to the control (Atg2A/2B-DKO + Myc Vec or DKO itself)?

Re: In response to your request, we conducted experiments to examine the palmitoylation levels of HA-ATG2A in Atg2A/2B-DKO +Atg2A Wt + Myc-Z11 cells. The results indicated that the palmitoylation level of ATG2A was weak under starvation conditions, but it could be significantly increased by overexpressing Z11. The data has been added as Fig. EV4C in the revised manuscript.

In Fig. 4, we only quantified cells with large GFP-LC3B puncta. Compared to Atg2A/B-DKO cells, Atg2A/2B-DKO cells transfected with Atg2A and Myc-Z11 showed a slight increase in small GFP-LC3B dots, possibly due to the presence of some unpalmitoylated Atg2A despite Z11 overexpression. Consistent with this, statistical analysis revealed that the formation of large GFP-LC3B puncta in Atg2A/2B-DKO cells transfected with Atg2A and Myc-Z11 is slightly

lower than in Atg2A/2B-DKO cells (Fig. 4D, E in the revised manuscript).

Minor concerns

Figure legends need to describe the experimental conditions for Fig. 3A and Fig. 6H.

Are cells starved?

Re: We apologize for this. The cells are not starved. In the revised manuscript, we have clarified the experimental conditions for Fig. 3A and Fig. 6H in the figure legends.

Referee #2:

The manuscript from Zheng et al. addresses the important question of how phospholipid flux across the membrane contact site between ER and phagophore mediated by ATG2, a conserved protein family of rod-like phospholipid transfer proteins, is regulated. They describe a novel depalmitoylation-dependent activation mechanism of ATG2, which is important for autophagy and lysosomal damage repair. In this study the authors report that, under nutrient rich conditions, ATG2A is post-translationally modified at three conserved cysteine residues at its C-terminus by the S-palmitoyltransferase ZDHHC11. This modification tethers the C-terminus of ATG2A to the ER membrane without interfering with the lipid transfer activity of the protein. During starvation, ATG2A is depalmitoylated by the depalmitoylase APT1, which releases the ATG2A C-terminus from the ER membrane and enables the interaction of ATG2A with the phagophore membrane - an important step during autophagy which yields in the formation of autophagosomes.

The manuscript describes data suggesting a new mechanism of how ATG2 binding to the phagophore might be regulated to control autophagosome biogenesis. Overall, the data are convincing and supports the conclusions of the authors. However, there are a few points that need to be addressed:

Re: We appreciate your positive evaluation of our study. We have revised the manuscript to address every point you pointed out.

Major comments:

1. It is required for good scientific practice and rigor that all Western blot analyses with quantitative interpretations are carefully quantified (n{greater than or equal to}3) and statistically analyzed.

Re: Following your suggestion, we carefully quantified and statistically analyzed the results of the Western blots.

2. Figure 2E: The authors argue that deletion of the first 42 residues of Z11 leads to disruption in the interaction between Z11 and ATG2A, because they see reduction in ATG2A band intensity. However, the amounts of immunoprecipitated Myc-Z11 vary between samples making it difficult to link the drop in ATG2A band intensity to impaired interaction between the two proteins or to the different amounts of Z11 proteins precipitated in this set-up. The authors need to (1) show less overexposed WBs and (2) quantify each ratio of ATG2A and the tested Z11 constructs to test for changes in interaction between ATG2A and the Z11 Δ 1-42 mutant. Additionally, to assess the biological outcome independently, the authors should check the localization of ATG2A and Z11 constructs in fluorescence microscopy using SEC61 labeled cells.

Re: Following your request, we conducted additional experiments to quantified the ratio of each ATG2A relative to its corresponding Z11s. The revised figures (Figs. 2E and EV2E) now include new, less exposed WBs and updated statistical data.

We examined the localization of ATG2A and the truncated Z11s in Sec61 β -labeled cells using fluorescence microscopy. The results indicate that the Z11-FL, Z11 (Δ 90-176), and Z11(1-251) constructs, which are capable of interacting with ATG2A, can form punctate structures with ATG2A. In contrast, the Z11 (Δ 1-42) truncation, which cannot interact with ATG2A, is unable to form such punctate structures. Due to the figure space, we include the images here for your reference. In Fig. EV2D, we identified that the large ATG2A puncta in these images are LDs.

3. Fig. 2G: Authors should show the IP blot of Z11 and Z11 Δ 1-42 constructs and quantify the ratio of Palm/Z11 or Z11 Δ 1-42 to quantitatively show the increase in ATG2A S-palmitoylation which they suggest in the main text.

Re: We have conducted additional experiments and included the IP blot for Z11 and Z11 Δ 1-42. However, since HA-ATG2A co-precipitated little Z11 Δ 1-42, it's challenging to quantify the ratio of Palm/ Z11 Δ 1-42. Instead, we quantified the ratio of Palm/HA-ATG2A to reflect ATG2A palmitoylation. These data are presented in Figs. 2G and EV2G in the revised manuscript.

4. Fig. 2H: Authors should show the IP blot for Myc-Z11 and quantify the ratio of Palm/Z11.

Re: We have conducted additional experiments and included the IP blot for Myc-Z11. To demonstrate the alteration in ATG2A S-palmitoylation, we quantified the ratio of Palm/HA-ATG2A. These data are presented in Figs. 2H and EV2H in the revised manuscript. And the ratio of Palm/Z11 is showed below.

5. Fig. 3B: Authors should show the IP blot for HA-APT1/2 and quantify the ratio of Palm/APT1 and APT2 and Palm/ATG2A.

Re: Additional experiments have been conducted, and we have showed the IP blot for HA-APT1/2. Furthermore, we quantified the ratio of Palm/Flag-ATG2A to demonstrate alterations in ATG2A S-palmitoylation. These data are presented in Figs. 3C and EV3A in the revised manuscript. And the ratio of Palm/APT1 and APT2 is showed below.

6. Fig. 3C: Authors should show the IP blot for APT1/2 and quantify the ratio of Palm/ATG2A.

Re: Done, showing in Figs. 3D and EV3B in the revised manuscript.

7. Fig. 3E: authors should show the IP blots for Myc-APT1 and Myc-APT1 (S119A).

Re: Done, presenting in Figs. 3E and EV3E in the revised manuscript.

8. The authors use a CoIP approach to study the interaction between ATG2A and ZDHHC11 and APT1/2. However, while this approach is consistent with direct interactions between ATG2A and Z11 or APT1/2, it does not demonstrate it, and it is formally possible that they interact indirectly via additional proteins. Demonstrating direct interactions between ATG2A and Z11 or APT1/2 would require an in vitro binding assay with purified proteins.

Re: Following your advice, we successfully expressed and purified GST-APT1/2 protein from *E. coli*, HA-Z11 protein from HEK293T cells, and Flag-ATG2A protein from HEK293S GnTI- cells. In vitro binding assays confirmed the direct interaction between ATG2A and Z11, as well as ATG2A and APT1/2. These data have been incorporated into the revised manuscript as Fig. 2C and Fig. 3B.

9. The authors investigate how the loss or overexpression of Z11 affects the starvation-induced degradation of p62 and the formation of LC3 puncta in starved cells. They further suggest that inhibition of the depalmitoylation of ATG2 replicates loss of ATG2 and results in the accumulation of large LC3 structures in the cytosol of starved cells. However, the authors only examine the effects of Z11 overexpression in these experimental set-ups. Since the authors identified APT1 as the major depalmitoylase for ATG2A, they should investigate how loss and overexpression of APT1 affects the degradation of p62 and the LC3-dot formation in the absence or presence of overexpressed Z11.

Re: In response to your request, we have conducted additional experiment to investigate the issues raised. Our results indicate that in the absence of APT1, EBSS treatment still leads to the degradation of p62, which can be mitigated by the overexpression of Z11. Furthermore, the overexpressing APT1 does not further enhance p62 degradation induced by EBSS treatment, and co-expression of Z11 effectively attenuated p62 degradation in these cells. Cell fluorescence microscopy and quantification of large GFP-LC3B puncta formation yield consistent results. These data have been incorporated as Fig. 4I, J and Fig. EV4E, F in the revised manuscript.

10. Fig EV2 A and B: The authors should include a control in which an ATG2 mutant is not anchored at the ER but still retains a connection to the growing phagophore and quantify the localization of ATG2 puncta with the ER in each condition.

Re: Following your suggestion, we conducted additional experiments incorporating ATG2A-3CS as a control. ATG2A-3CS is not anchored to the ER; however, it retains the capacity to interact with the phagophore. Our findings indicate that all these variants were comparably distributed across ER fraction. The quantification data have been added. These data have been incorporated into the revised manuscript as Fig. EV5A-D.

Minor comments:

1. Some figures throughout the manuscript are incompletely labeled and insufficiently described in the figure legends. For example, in Fig.1A, B, C authors use HAM (hydroxylamine), without describing what HAM is in the main text and figure legend. In Figure 1D, authors use palmitic acid, however, they do not mention it in the figure legend. Carefully describing all sample treatments in the figure legends throughout the manuscript will be beneficial for the reader.

Re: We appreciate your feedback. We have reviewed all the figure legends and made modifications to several of them by providing necessary information regarding the experiments.

2. Figure 1A, 1B, 1C, 1E, 1F,1H,1I, 2A, 3D, 3G, 3H,6I: authors should add the Input blots.

Re: Thank you for bringing this to our attention. In accordance with your request, we have incorporated the Input blots into all the relevant figures.

3. Fig. 2B: Authors should correctly label the bands for Z11 and Z6 in the shown Western Blot image.

Re: We appreciate your attention to detail, and this has been corrected accordingly.

4. Figure 2D: Can the authors specify if the cells were grown in full media and add this information to the figure legend?

Re: Yes, they were indeed cultured in complete media. This detail has been incorporated into the figure legend of the revised manuscript. Thank you.

5. Fig. 2F: authors should show the IP blot for Z11 constructs.

Re: We have included the IP blot for Z11 and provided a quantification of HA-ATG2A palmitoylation levels as shown in Figs 2F and EV2F of the revised manuscript.

6. Fig. 4E: a negative control for LC3-dot formation is missing. The authors should include as a control cells grown in nutrient rich conditions.

Re: Thank you for your suggestion. We have added the control cells in Fig. 4F of the revised manuscript.

7. The authors show that ATG2A S-palmitoylation affects the interaction between ATG9 and ATG2A. Is ATG9 required for ATG2A depalmitoylation?

Re: This is an interesting question. To test the possibility, we examined ATG2A palmitoylation in WT and ATG9A-KO HeLa cells (PMID 27934868) under starvation conditions. The results indicate that in the absence of ATG9A, EBSS treatment reduced the palmitoylation level of ATG2A similarly to WT cells. Since no significant changes were observed and this data is not closely related to the main study, we included it here for your reference instead of in the manuscript.

Referee #3:

ATG2A is a lipid transfer protein that bridges ER and phagophore and transfers lipids from ER to phagophore for membrane elongation to build up an autophagosome. Although the lipid transfer activity of ATG2A has been well characterized, the regulation mechanism of the activity remains largely unknown. In this manuscript, the authors found that ATG2A is palmitoylated at Cys1704, 1713, and 1714 by ER-resident palmitoyltransferase ZDHHC11 and depalmitoylated by depalmitoylase APT1. ZDHHC11 directly bound ATG2A and showed colocalization with ATG2A. Expression of ATG2A with mutations at palmitoylation sites enhanced autophagy activity but the same mutations did not enhance the lipid transfer activity of purified ATG2A. Further analyses showed that palmitoylation of ATG2A made the C-terminal

region of ATG2A to bind to ER and thereby inhibit the ER-phagophore bridging. Based on these data, the authors concluded that the lipid transfer activity of ATG2A is negatively regulated by palmitoylation by making the C-terminus of ATG2A to bind to ER and activated by depalmitoylation which enables ATG2A to bridge between ER and phagophore and transfer lipids from ER to phagophore for autophagosome formation.

ATG2-mediated lipid transfer is one of the critical steps in autophagosome formation and its regulation by palmitoylation is novel and interesting. The experiments are well designed and the obtained results are clear and convincing. Thus this manuscript will be attractive to the broad readers of EMBO Journal. However, one of the major conclusions is inconsistent with previous reports and need to be resolved.

Re: We appreciate your positive feedback on our study and are glad you enjoyed the story. We conducted additional experiment to address the key points you raised.

Major points

1) The main conclusion that ATG2A localizes predominantly to ER by S-palmitoylation is not consistent with previous reports showing that ATG2A mainly localizes at lipid droplets and phagophores, but not at ER (PMID 22219374, 29113029). In Fig. 2D, the authors observed that ATG2A formed punctual structures with ZDHHC11; however, in contrast to the ER pattern of ZDHHC11, ATG2A did not show such pattern. The punctual structures seem to be lipid droplets rather than ER. In Fig. EV2B, the authors compared the localization of ATG2A and Sec61beta, an ER-marker, which also showed that ATG2A colocalizes with Sec61beta as puncta but not as ER pattern. Analyze the colocalization of ATG2A, ZDHHC11, lipid droplet marker, and ER marker and clearly indicate whether the colocalization site of ATG2A and ZDHHC11 is ER or lipid droplets. If the site corresponds to lipid droplets, reconsider the model shown in Fig. 6L.

Re: Thank you for your professional vision and comments.

It is well established that ATG2 functions as a non-vesicular lipid transport between the ER and phagophore by localizing at their contact sites. Accumulating evidence indicates that ATG2 possesses two membrane-binding domains located in its N- and C-terminal regions respectively (PMID 30952800, PMID 22219374, 29113029). In two published studies (PMID 22219374, 29113029), Dr. Mizushima and colleagues elegantly demonstrated the localization of ATG2A to LDs. They also identified that ATG2A localizes to both the isolation membrane and LDs through an amphipathic helix presented in the C-terminal region (AH, aa 1750-1767). In this study, we propose that ATG2A associated with the ER not only because it directly interacts with ZDHHC11, an ER-localized transmembrane palmitoyltransferase (new data from protein-binding assays using purified proteins, Fig. 2B) but also based on subcellular fractionation assays showing the distribution of ATG2A lacking the amphipathic helix (ATG2A- Δ AH). Notably, even the unpalmitoylatable mutant ATG2A-3CS- Δ AH exhibited equal distribution within the ER fraction. These data strongly suggest that while ATG2A interacts with either phagophore or LD through its C-terminal region (AH), it associates with the ER via its N-terminal membrane-binding domain.

Because ATG2A does not possess an integral transmembrane domain, akin to that of Atg14, it is challenging to observe a nice ER-localization pattern like ZDHHC11 or Sec61 β . Additionally, the strong affinity of its C-terminal region for the unique phospholipid monolayer of LDs may lead to an oversight regarding the association of the N-terminal region of ATG2A with the ER under microscopic examination.

In accordance with your valuable suggestion, we conducted further experiments to elucidate the colocalization of ATG2A-ZDHHC11 puncta with both Sec61 β and LDs. Our observation confirmed colocalization between these puncta and Sec61 β , as well as established localization of ATG2A to LDs, no colocalization was detected between ATG2A-ZDHHC11 puncta and LDs. These

data provide additional support for our proposal and have been incorporated into Fig. EV2C and EV2D in the revised manuscript.

2) Recently, a Halo-tag assay (PMID 35938926) was developed to more accurately quantify the autophagic flux compared with conventional methods used in this manuscript. Perform Halo-tag assay to further confirm the effect on the autophagic flux.

Re: This is an excellent suggestion. Following your recommendation, we have conducted the Halo-tag assays and obtained consistent clear results. These data have been added as Fig. 4H and EV4D in the revised manuscript. Thank you.

Minor points

1) As an example of S-palmitoylation in the regulation of autophagy, a recent manuscript (PMID 39362856) should also be cited in the Introduction.

Re: Thank you for the reminder. We have now included this reference in the revised manuscript's Introduction section.

2) As to Fig. 1A-C, add a brief explanation of the ABE assay in the main text and of Palm in the legend.

Re: Thank you for your valuable advice. We have incorporated a concise description of the ABE assay in the main text and information regarding 'Palm' in the figure legend.

3) In Fig. 1D, cells that do not express Flag-ATG2A and have been treated with 17-ODYA should be added as a negative control.

Re: Cells lacking Flag-ATG2A expression and subjected to 17-ODYA treatment have now been added as a negative control, as illustrated in Fig. 1D.

4) The authors discussed that "S-palmitoylation only affects the interaction between ATG2A and Atg9, but not with WIPI4. This may be due to the fact that the ATG9-binding domain (aa 1760-1779) on ATG2A is closer to the S-palmitoylation sites than the WIPI4-binding domain (aa 1358-1404)" in page 16. However, based on

the model proposed by the authors, this reviewer thinks that S-palmitoylation anchors the C-terminus of Atg2 to the ER, making it unable to interact with the autophagic membrane, and therefore, unable to interact with ATG9, a transmembrane protein that does not localize to the ER but to the autophagic membrane (in contrast, WIPI4 is not a transmembrane protein and thus can move to the ER surface and binds to ER-anchored ATG2A). Reconsider the discussion.

Re: We sincerely appreciate your thoughtful suggestions. After careful consideration, we fully align with your perspective, which also corresponds to our findings. Following your advice, we have revised and enhanced our discussion regarding this issue in the Discussion section of our manuscript.

Dear Prof. Liu,

We have now received re-review reports from two referees, which I have included below. As you will see, you have addressed their concerns satisfactorily; however, I would like you to consider addressing their remaining points in the discussion section. Before I can finally accept the manuscript, there are some remaining editorial points which need to be addressed. In this regard would you please:

- move the Data Availability section to before the Acknowledgments section, rename Table EV2 as Dataset EV1 (source file name, title in our online submission system, and manuscript callouts); Table EV4 should then be renamed Table EV3 in a similar manner,
- combine source data folders for EV figures into one zipped folder,
- rename 'Methods and Protocols' as 'Methods', and provide the exact p values in the legends of figures 1D, 3H, 4E, G; 5D, G, I, K; 6B, D, F, J, L; EV1 A, B, C, D, G, H; EV2 A, E, F, G, H; EV3 A, B, D, E, F, J; and EV4 A, B, D, E, F.

We include a synopsis of the paper (see <http://emboj.embopress.org/>). Please provide me with a general summary image, a two sentence statement and 3-5 bullet points that capture the key findings of the paper.

I am looking forward to receiving your revised manuscript.

EMBO Press is an editorially independent publishing platform for the development of EMBO scientific publications.

Best wishes,

William

William Teale, PhD
Editor
The EMBO Journal
w.teale@embojournal.org

We realize that it is difficult to revise to a specific deadline. In the interest of protecting the conceptual advance provided by the work, we recommend a revision within 3 months (18th May 2025). Please discuss the revision progress ahead of this time with the editor if you require more time to complete the revisions. Use the link below to submit your revision:

Referee #1:

Zheng and colleagues reveal that S-palmitoylation regulates the function of the C-terminus of ATG2A controlled by ZDHHC11 and APT1, allowing localization changes from the ER to the phagophore under the starved condition. In their revised manuscript, the authors responded to all points stated. However, information on how Atg2A is regulated to be less interactive with ZDHHC11 is missing (authors showed APT1 does not play a role); the paper is improved and clearer. Localization issues also raised by other authors are concerning; however, if an over-expression system conducts experiments, it will all depend on cell types and expression system; it is a common issue working with mammalian cells; thus, unless proteins could be expressed at the endogenous level with endogenous promoters, no point in discussing further at this point. No further significant concerns. It is exciting to see another S-acylation regulation in autophagy.

Minor concerns

- 1) A typo on page 13, where "ZDHH11" should be corrected to "ZDHHC11".
- 2) A suggestion to modify the Y-axis labels on figures EV1 and EV3 for clarity, proposing to change "Palm vs. HA-Atg2" to "Palm on HA-Atg2" or a similar phrasing.

Referee #3:

The authors have addressed all of my concerns.

Referee #1:

Zheng and colleagues reveal that S-palmitoylation regulates the function of the C-terminus of ATG2A controlled by ZDHHC11 and APT1, allowing localization changes from the ER to the phagophore under the starved condition. In their revised manuscript, the authors responded to all points stated. However, information on how Atg2A is regulated to be less interactive with ZDHHC11 is missing (authors showed APT1 does not play a role); the paper is improved and clearer. Localization issues also raised by other authors are concerning; however, if an over-expression system conducts experiments, it will all depend on cell types and expression system; it is a common issue working with mammalian cells; thus, unless proteins could be expressed at the endogenous level with endogenous promoters, no point in discussing further at this point.

No further significant concerns. It is exciting to see another S-acylation regulation in autophagy.

Re: We sincerely appreciate your constructive and supportive comments on our revised manuscript. Regarding the diminished interaction between ATG2A and ZDHHC11 in starved cells, the precise mechanism remains unclear at this time. Considering that the S-palmitoylation level of ATG2A is reduced under conditions of glucose and serum starvation, as well as following Torin 1 treatment and lysosomal damage, we hypothesize a potential involvement of mTOR activity in the process. Alterations in modifications to either ATG2A or ZDHHC11 directly or indirectly by mTOR may contribute to the reduction in ATG2A-ZDHHC11 interaction. We have addressed this in the Discussion section and are currently investigating this possibility in an ongoing study.

Due to the lack of commercially available antibodies suitable for immunostaining endogenous ZDHHC11 and ATG2A, we employed an overexpression system to examine their subcellular localization. This approach is currently the most widely adopted method for determining the localization of these proteins. In addition, the localization patterns of ATG2A and ZDHHC11 observed in our experiments align with previously reported findings (PMID:

39143266; 30952800; 31412244), suggesting that the overexpression may have a limited impact. We hope this explanation meets your approval.

Minor concerns

1) A typo on page 13, where "ZDHH11" should be corrected to "ZDHHC11".

Re: It has been corrected. Thank you.

2) A suggestion to modify the Y-axis labels on figures EV1 and EV3 for clarity, proposing to change "Palm vs. HA-Atg2" to "Palm on HA-Atg2" or a similar phrasing.

Re: We have changed the Y-axis labels following your kind suggestion.

Referee #3:

The authors have addressed all of my concerns.

Re: We sincerely appreciate your invaluable insights and constructive suggestions, which have significantly enhanced the quality of our work.

Dear Prof. Liu,

I am pleased to inform you that your manuscript has been accepted for publication in the EMBO Journal.

Congratulations to you and your team on the publication of this work!

Please provide a file of your synopsis image that exactly 550 pixels wide x 200-600 pixels high.

Yours sincerely,

William Teale

William Teale, PhD
Editor
The EMBO Journal
w.teale@embojournal.org
